# MTSAM: Multi-Task Fine-Tuning for Segment Anything Model

**Xuehao Wang[1], Zhan Zhuang[1,2], Feiyang Ye[1,3], Yu Zhang[1,*]**
[1]Southern University of Science and Technology
[2]City University of Hong Kong
[3]University of Technology Sydney
{xuehaowangfi,feiyang.ye.uts,yu.zhang.ust}@gmail.com
12250063@mail.sustech.edu.cn

## Abstract

The Segment Anything Model (SAM), with its remarkable zero-shot capability, has the potential to be a foundation model for multi-task learning. However, adopting SAM to multi-task learning faces two challenges: (a) SAM has difficulty generating task-specific outputs with different channel numbers, and (b) how to fine-tune SAM to adapt multiple downstream tasks simultaneously remains unexplored. To address these two challenges, in this paper, we propose the **Multi-T**ask **SAM** (**MTSAM**) framework, which enables SAM to work as a foundation model for multi-task learning. MTSAM modifies SAM's architecture by removing the prompt encoder and implementing task-specific no-mask embeddings and mask decoders, enabling the generation of task-specific outputs. Furthermore, we introduce **T**ensorized l**o**w-**R**ank **A**daptation (**ToRA**) to perform multi-task fine-tuning on SAM. Specifically, ToRA injects an update parameter tensor into each layer of the encoder in SAM and leverages a low-rank tensor decomposition method to incorporate both task-shared and task-specific information. Extensive experiments conducted on benchmark datasets substantiate the efficacy of MTSAM in enhancing the performance of multi-task learning. Our code is available at https://github.com/XuehaoWangFi/MTSAM.

## 1 Introduction

Empowered by large-scale datasets and computational advancements, large foundation models have revolutionized natural language processing and multi-modal learning, exhibiting remarkable zero-shot capabilities (Radford et al., 2018; 2019; Kenton & Toutanova, 2019; Lewis et al., 2020; Brown et al., 2020; Radford et al., 2021; Wei et al., 2023; Jiang et al., 2024). Recently, the Segment Anything Model (SAM) (Kirillov et al., 2023), a foundation model in computer vision for image segmentation, achieves exceptional zero-shot performance through training on a large-scale dataset of 11 million samples. Efforts have been dedicated to expanding the zero-shot capability of SAM to various tasks, including high-quality segmentation (Ke et al., 2023), 3D reconstruction (Cen et al., 2023), object tracking (Yang et al., 2023), medical image processing (Ma et al., 2024; Huang et al., 2024), personalize segmentation (Zhang et al., 2023), and remote sensing (Shankar et al., 2023).

Though SAM has achieved remarkable performance in diverse tasks in previous studies, those studies only adopt SAM to a specific downstream task by single-task learning, while overlooking the potential of employing SAM as the foundation model for multi-task learning. In many real-world computer vision applications, there is usually more than one task that can be considered simultaneously, such as depth estimation and surface normal estimation tasks in dense scene understanding. Previous works on multi-task learning (Misra et al., 2016; Liu et al., 2019; Ye & Xu, 2022; 2023; Zamir et al., 2018; Liu et al., 2022) have shown that these tasks are relevant and can benefit each other during the training process. This insight motivates us to adopt SAM as a foundation model for multi-tasking learning to enhance the performance of different tasks. However, adopting SAM

---

*Corresponding author.

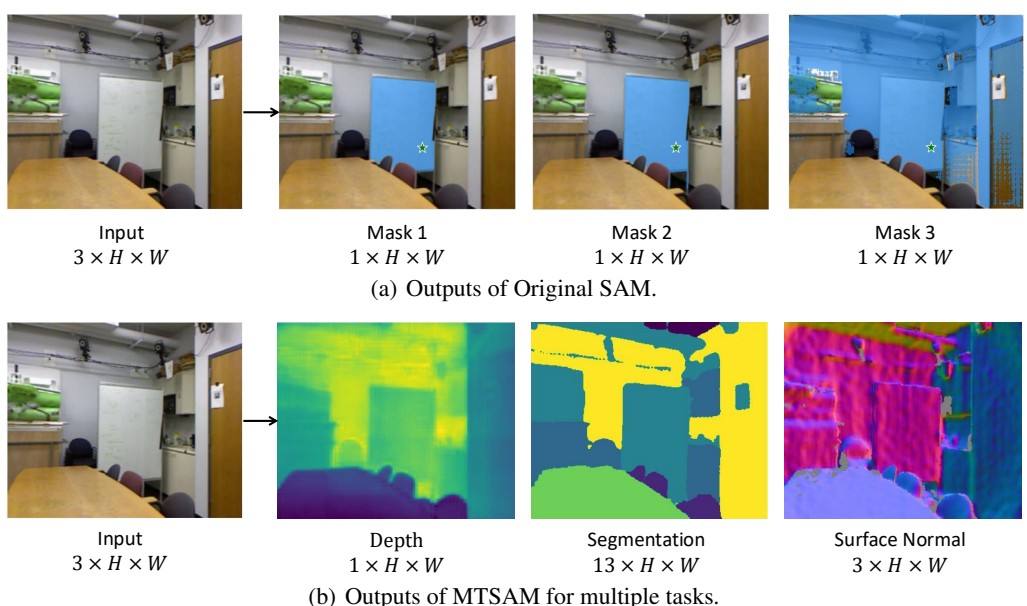

(a) Outputs of Original SAM.

(b) Outputs of MTSAM for multiple tasks.

Figure 1: Comparison between the outputs of (a) the original SAM and (b) the MTSAM proposed in this paper. The original SAM generates segmentation results at three distinct levels, all featuring an identical number of channels. In contrast, the MTSAM can simultaneously produce outputs for multiple tasks, utilizing varying numbers of channels.

to multi-task learning presents two challenges: (a) *how to generate outputs with varying dimensions for each task by SAM*; and (b) *how to fine-tune SAM to adapt multiple tasks simultaneously*.

To tackle the above challenges, we propose the Multi-Task SAM (MTSAM) framework to leverage the rich knowledge from SAM for multi-task learning. Our approach includes two key modifications: (a) adapting the architecture of SAM to accommodate varying channel numbers for each task and (b) introducing a novel multi-task parameter-efficient fine-tuning method named Tensorized low-Rank Adaption (ToRA) for fine-tuning the SAM encoder.

Specifically, we incorporate task embeddings and remove the prompt encoder from the mask decoder, enabling the generation of the outputs with dimensions tailored to each task. As illustrated in Figure 1, this modification enhances the flexibility of SAM, allowing it to adapt to various tasks within a unified architecture. For multi-task fine-tuning, ToRA injects an update parameter tensor into each layer of the SAM encoder, with each slice of the tensor serving as an update parameter matrix for the corresponding task. Inspired by Low-Rank Adaptation (LoRA) (Hu et al., 2021), we assume a low-rank structure for each update parameter tensor and apply a low-rank tensor decomposition to capture both task-shared and task-specific information. Theoretically, we prove that ToRA's expressive power in multi-task learning surpasses that of LoRA. ToRA exhibits superior parameter efficiency, with sublinear growth in learnable parameters with respect to the number of tasks, in contrast to the linear growth in the original LoRA method when applied directly to multiple tasks.

The main contributions of this paper are three-fold. (i) We propose MTSAM, a novel multi-task learning framework that extends the capabilities of SAM to perform multi-task learning. Specifically, we modify the original architecture of SAM by removing the prompt encoder and adding task embeddings. This modification enhances the flexibility of the original SAM. (ii) We introduce ToRA, a novel multi-task PEFT method, that applies low-rank decomposition to the update parameter tensor, effectively learning both task-shared and task-specific information simultaneously, with theoretical analysis of its strong expressive power. (iii) We conduct comprehensive experiments on benchmark datasets, demonstrating the exceptional performance of the MTSAM framework.

## 2 RELATED WORKS

**Application of Segment Anything Model (SAM).** The remarkable zero-shot generalization ability exhibited by SAM showcases its immense potential for both research and industrial applications.

This potential has captured the attention of researchers, leading to numerous attempts to explore and harness its capabilities for various downstream tasks, including all-purpose matching (Liu et al., 2023b), high-quality segmentation (Ke et al., 2023), 3D reconstruction (Cen et al., 2023), object tracking (Yang et al., 2023), medical image processing (Ma et al., 2024; Huang et al., 2024), personalize segmentation (Zhang et al., 2023), and remote sensing (Shankar et al., 2023). In contrast to those modifications targeted at a single downstream task, the proposed MTSAM aims to simultaneously learn multiple downstream tasks and extract semantic knowledge from SAM to enhance the performance of multi-task learning.

**Parameter-Efficient Fine-Tuning (PEFT).** To address the parameter and computational efficiency concerns during fine-tuning of large-scale pre-trained foundation models, various PEFT methods have been proposed, including adapter-based methods (Houlsby et al., 2019; Lin et al., 2020), prompt tuning methods (Li & Liang, 2021; Lester et al., 2021; Chen et al., 2024; Jiang et al., 2023), and LoRA-based methods (Zhong et al., 2023; Kopiczko et al., 2023; Wu et al., 2023; Hu et al., 2021; Ding et al., 2023; Valipour et al., 2023; Guo et al., 2025). Specially, LoRA (Hu et al., 2021) introduces trainable low-rank matrices into transformer layers to approximate update parameter matrix, Conv-LoRA (Zhong et al., 2023) inserts Mixture of Experts (MoE) (Jacobs et al., 1991) inside the bottleneck of LoRA, VeRA (Kopiczko et al., 2023) fixes the low-rank matrices and only tunes two vectors, MoLE (Wu et al., 2023) directly uses multiple LoRA which combined by a gating function, SoRA (Ding et al., 2023) enables dynamic adjustments of the rank by using a gate unit, and DyLoRA (Valipour et al., 2023) follows the idea of nested dropout to train LoRA in a wide range of ranks. Those methods achieve competitive performance and high parameter efficiency in single-task fine-tuning. However, those methods are not suitable for multi-task learning settings, since they do not consider shared information between multiple tasks. In contrast, the proposed ToRA method can leverage task-shared information to enhance fine-tuning performance across various tasks.

**Multi-Task Learning (MTL).** As a widely used paradigm, MTL aims to improve the average performance of a model by simultaneously learning multiple downstream tasks. To enhance the efficacy of learning multiple tasks simultaneously, some studies focus on decoupling task-shared and task-specific information through manual design (Misra et al., 2016; Liu et al., 2019; Ye & Xu, 2022; 2023; Gao et al., 2019; Lin et al., 2024b;a) or automatic architecture learning (Guo et al., 2020; Huang et al., 2018; Raychaudhuri et al., 2022; Sun et al., 2020). Other approaches propose balancing the losses or gradients of different tasks during training to avoid conflicts between them (Chen et al., 2018; Yu et al., 2020; Liu et al., 2021b;a; Ye et al., 2021; Navon et al., 2022; Ye et al., 2023; Lin et al., 2023). Additionally, some works employ task grouping techniques to select related tasks for joint model training (Fifty et al., 2021; Song et al., 2022; Standley et al., 2020; Zamir et al., 2018). With the impressive generalization capability of large-scale pre-trained foundation models on downstream tasks, various multi-task parameter-efficient fine-tuning methods (Liu et al., 2022; 2023a) have been proposed. For example, Polyhistor (Liu et al., 2022) designs a lightweight hypernetworks for hierarchical vision transformer, and HiPro (Liu et al., 2023a) uses hierarchical prompt tuning to adapt pre-trained vision-language models. Different from the previous works on multi-task learning, we leverage the powerful SAM and propose the MTSAM framework which uses a novel method ToRA to fine-tune the encoder.

## 3 METHODOLOGY

In this section, we introduce the proposed MTSAM framework and ToRA method.

### 3.1 PRELIMINARIES

**SAM.** The original SAM consists of three main modules: a heavyweight image encoder, a prompt encoder, and a lightweight mask decoder. Given an image $I \in \mathbb{R}^{3 \times H \times W}$, where $H$ and $W$ denote the height and width of the image $I$, respectively. SAM first utilizes the image encoder $E_I$ to extract image features $F_I \in \mathbb{R}^{D \times \frac{H}{16} \times \frac{W}{16}}$ as

$$F_I = E_I(I), \tag{1}$$

where $D$ denotes the dimension of the hidden state. Then the prompt encoder, which consists of a dense mask encoder $E_M$ and a sparse prompt encoder $E_P$, encodes dense masks $M \in \mathbb{R}^{1 \times \frac{H}{4} \times \frac{W}{4}}$ and different types of sparse prompts $P$ (i.e., points, box, and text) into mask features $F_M$ and

prompt features $F_P$ as

$$F_M = E_M(M), \; F_P = E_P(P), \tag{2}$$

where $F_M, F_P \in \mathbb{R}^{D \times \frac{H}{16} \times \frac{W}{16}}$. After that, the image features are summed with the mask features, and the prompt features are concatenated with some learnable prompt features $F_L$. Finally, the mask decoder $D_M$ will predict the final segmentation mask output $O \in \mathbb{R}^{3 \times \frac{H}{4} \times \frac{W}{4}}$ by performing attention-based feature interactions on image features and prompt features as

$$O = D_M(F_I + F_M, [F_L, F_P]). \tag{3}$$

**LoRA.** The LoRA method (Hu et al., 2022) assumes that each update parameter matrix has a low intrinsic rank and fine-tunes them by freezing the pre-trained model. Formally, for a given task, LoRA parameterizes an update parameter matrix $\Delta W \in \mathbb{R}^{d \times k}$ corresponding to a pre-trained parameter matrix $W_0 \in \mathbb{R}^{d \times k}$ by the product of two low-rank matrices, i.e., $\Delta W = BA$, where $B \in \mathbb{R}^{d \times r}$ and $A \in \mathbb{R}^{r \times k}$ with $r \ll \min(d, k)$. Thus, for an input $x$, the output $h$ can be calculated by

$$h = W'x = W_0 x + \Delta W x = W_0 x + BAx, \tag{4}$$

where the $W' \in \mathbb{R}^{d \times k}$ denotes the parameter matrix after the update. LoRA has been proven as an efficient and effective approach in fine-tuning large pre-trained models for specific downstream tasks. Therefore, we consider LoRA as an important baseline method in our experiments. There are two different approaches to directly applying LoRA to the multi-task fine-tuning setting for SAM. One approach uses a hard parameter sharing strategy, where all tasks use one shared LoRA matrix $\Delta W$, and we call it LoRA-HPS. However, this hard parameter-sharing strategy may lead to imbalanced performance on all the tasks due to the competition among tasks for the shared LoRA (Zhang & Yang, 2022). Another approach is to train a task-specific LoRA for each task and hence each task $t$ uses its own $\Delta W_t$, and we call it LoRA-STL. However, this approach cannot harness the inter-task shared information necessary for fine-tuning across multiple tasks.

## 3.2 ARCHITECTURE

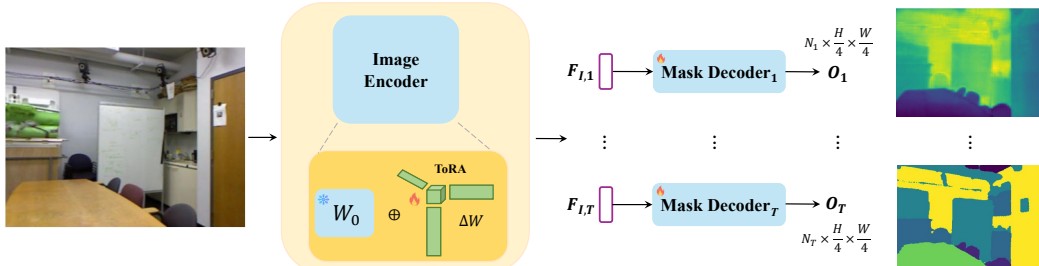

Figure 2: An overview of the proposed MTSAM. The proposed ToRA is used to fine-tune the heavyweight image encoder and generate task-specific image embeddings for each task. MTSAM does not utilize the prompt encoder of the original SAM and modifies the mask decoder of SAM to generate outputs with varying numbers of output channels (denoted by $N_i$ for task $i$).

Despite the tremendous potential exhibited by SAM as a fundamental visual model, its reliance on prompt-guided mask generation presents challenges in achieving end-to-end adaptability to downstream tasks with varying numbers of output channels. Therefore, we propose the MTSAM to enable end-to-end multi-task fine-tuning for SAM.

As shown in Figure 2, MTSAM follows the standard encoder-decoder architecture of SAM, including a heavyweight image encoder and several task-specific lightweight mask decoders. Different from SAM, MTSAM removes the prompt encoder in the original SAM and modifies the architecture of the mask decoder.

To fully leverage the rich semantic knowledge acquired by SAM during pre-training, we froze the pre-trained parameters in the heavyweight image encoder and employ multi-task fine-tuning, which will be introduced in the next section, to update parameter tensors in the self-attention module (i.e., the query, key, and the value) of the image encoder. Moreover, we perform fine-tuning on the scale and bias parameters within the layer normalization layers of the image encoder.

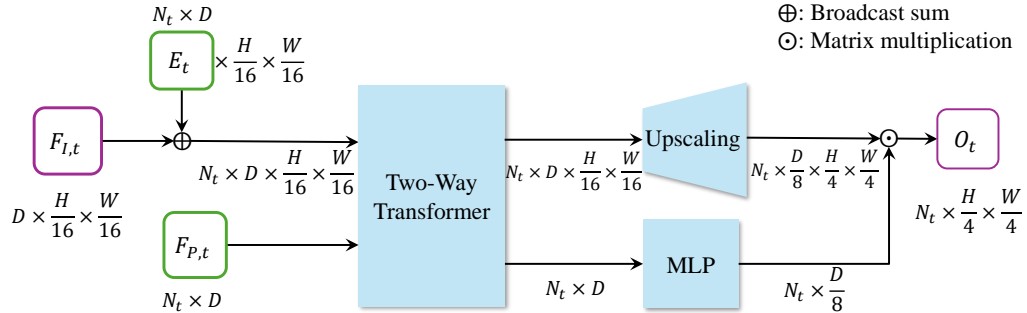

Figure 3: The mask decoder of MTSAM for task $t$, which has $N_t$ output channels.

To adapt the entire model to various tasks, MTSAM introduces separate mask decoders for each task, generating task-specific outputs. Specifically, as detailed in Figure 3, we introduce trainable task embeddings with distinct numbers of output channels and consequently discard the dense mask encoder and the sparse prompt encoder in SAM. Formally, the task embeddings $E_t \in \mathbb{R}^{N_t \times D}$ for task $t$, where $N_t$ denotes the number of output channels for task $t$, is expanded to $E'_t \in \mathbb{R}^{N_t \times D \times \frac{H}{16} \times \frac{W}{16}}$ by copying $E_t$ for $\frac{H}{16} \times \frac{W}{16}$ times. Then, we perform the broadcast sum between the image embedding $F_{I,t} \in \mathbb{R}^{D \times \frac{H}{16} \times \frac{W}{16}}$ of task $t$ and the expanded task embeddings $E'_t$, i.e., $(F_{I,t} \oplus E'_t) \in \mathbb{R}^{N_t \times D \times \frac{H}{16} \times \frac{W}{16}}$. Another input to the mask decoder is a learnable token $F_{P,t} \in \mathbb{R}^{N_t \times D}$. Then the two types of input are fed into a two-way Transformer (Kirillov et al., 2023) and the outputs consist of a hidden image feature representation and a hidden token feature, which are fed into an upscaling layer and an MLP layer, respectively. In particular, the upscaling layer uses the transposed convolution operator. Thus, the decoder generates the prediction $O_t \in \mathbb{R}^{N_t \times \frac{H}{4} \times \frac{W}{4}}$ for task $t$ as

$$O_t = D_t(F_{I,t} \oplus E'_t, F_{P,t}). \tag{5}$$

### 3.3 TENSORIZED LOW-RANK ADAPTATION FOR MULTI-TASK FINE-TUNING

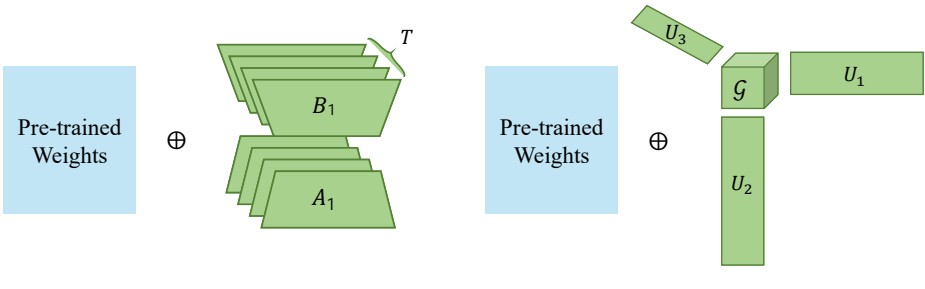

(a) LoRA for multiple tasks.     (b) ToRA for multiple tasks.

Figure 4: Comparison between (a) LoRA and (b) ToRA. LoRA uses separate low-rank matrices for the update parameter matrix of each task, while ToRA aggregates the update parameter matrices of all the tasks into an update parameter tensor and applies low-rank tensor decomposition.

To efficiently fine-tune the computationally intensive image encoder in MTSAM, we propose Tensorized low-Rank Adaptation (ToRA), a novel parameter-efficient multi-task fine-tuning method. The comparison between LoRA and ToRA is shown in Figure 4.

Suppose we are given $T$ tasks. For simplicity, we consider the case where each task involves fine-tuning only one layer using a PEFT approach, although this can be easily extend to multiple layers. For task $t$, we denote its update parameter matrix as $\Delta W_t \in \mathbb{R}^{d \times k}$. Consider all the $T$ tasks, it is natural to aggregate the update parameter matrices of all the tasks into an update parameter tensor $\Delta \mathbf{W} = \{\Delta W_1, \dots, \Delta W_T\} \in \mathbb{R}^{d \times k \times T}$. Inspired by LoRA, we impose a low-rank assumption on $\Delta \mathbf{W}$. This assumption is plausible due to both inter-task relatedness and intra-task low rank in each $\Delta W_t$, as observed in LoRA. Specifically, since different tasks in multi-task learning are typically assumed to be related, the update parameter matrices $\{\Delta W_t\}_{t=1}^{T}$ may be correlated, making $\Delta \mathbf{W}$ likely to be low-rank along the task axis (i.e., the last axis). In this sense, this low-rank assumption

on $\Delta \mathbf{W}$ could be viewed as a generalization of the low-rank assumption on the parameter matrix of linear models (Zhang & Yang, 2022).

To achieve a low-rank $\Delta \mathbf{W}$, we parameterize it via tensor decomposition (Papalexakis et al., 2016), which is a technique to decompose a tensor into several low-rank factors. There are several tensor decomposition methods (e.g., CP decomposition (Hitchcock, 1927; Carroll & Chang, 1970) and Tucker decomposition (Tucker, 1966)) and we choose the Tucker decomposition as it has a good representation ability (Papalexakis et al., 2016). Specifically, we decompose the three-mode update parameter tensor $\Delta \mathbf{W} \in \mathbb{R}^{d \times k \times T}$ into a core tensor $\mathcal{G} \in \mathbb{R}^{p \times q \times v}$ and three factor matrices $U_1 \in \mathbb{R}^{d \times p}$, $U_2 \in \mathbb{R}^{k \times q}$ and $U_3 \in \mathbb{R}^{T \times v}$, where $p$, $q$, and $v$ denote the dimensions of factor matrices, typically $p, q, v \ll \min(d, k)$. Formally, this can be expressed as

$$\Delta \mathbf{W} = \mathcal{G} \times_1 U_1 \times_2 U_2 \times_3 U_3, \tag{6}$$

where $\times_n$ denotes the $n$-mode product. Accordingly, the $(i, j, t)$-th entry in $\Delta \mathbf{W}$ can be written as

$$\Delta \mathbf{W}(i, j, t) = \sum_{m=1}^{p} \sum_{n=1}^{q} \sum_{l=1}^{v} \mathcal{G}(m, n, l) U_1(i, m) U_2(j, n) U_3(t, l), \tag{7}$$

where $i \in \{1, 2, \ldots, d\}$, $j \in \{1, 2, \ldots, k\}$, and $t \in \{1, 2, \ldots, T\}$ denote the indices of three mode, respectively. For the three-mode update parameter tensor $\Delta \mathbf{W}$, the first mode represents the output feature dimension, the second mode denotes the input feature dimension, and the third mode is for the task dimension. Hence, according to Tucker decomposition (Tucker, 1966), $U_1$ and $U_2$ reflect the main subspace variation of task-shared information corresponding to the first two modes in $\Delta \mathbf{W}$, while $U_3$ reflects the task-specific subspace structure corresponding to the last mode of $\Delta \mathbf{W}$. Hence, through the Tucker tensor decomposition, the ToRA method could capture both task-shared and task-specific information.

**Initialization.** For the proposed ToRA method, the core tensor $\mathcal{G}$ is initialized as $\mathbf{0}$, while factor matrices $U_1$, $U_2$, and $U_3$ are randomly initialized from the standard Gaussian distribution. Thus, for each task $t$, the update parameter matrix $\Delta W_t$ is $\mathbf{0}$ at the beginning of training.

**Training and inference.** During training, we utilize Eq. (6) to obtain the update parameter tensor $\Delta \mathbf{W}$ based on $U_1$, $U_2$, $U_3$, and $\mathcal{G}$ and employ $h = W_t' x = W_0 x + \Delta \mathbf{W}(:, :, t)x$ on the forward process for task $t$. During the back-propagation process, we freeze the pre-trained matrix $W_0$ and only update $U_1$, $U_2$, $U_3$, and $\mathcal{G}$. During inference, we can store the updated parameter matrix of task $t$ as $W_t = W_0 + \Delta \mathbf{W}(:, :, t)$. Thus, there is no additional latency introduced during inference.

**Parameter complexity.** We present a comparison of parameter complexity between LoRA and the proposed ToRA under the multi-task learning setting. To fine-tune the pre-trained matrix $W_0 \in \mathbb{R}^{d \times k}$, the LoRA method decomposes each update parameter matrix $\Delta W_t$ as $B_t A_t$, where $B_t \in \mathbb{R}^{d \times r}$ and $A_t \in \mathbb{R}^{r \times k}$. Therefore, for $T$ tasks, the parameter complexity of LoRA is $\mathcal{O}(Trd + Trk)$. For the proposed ToRA method, we decompose the update parameter tensor $\Delta \mathbf{W} \in \mathbb{R}^{d \times k \times T}$ as $\mathcal{G} \times_1 U_1 \times_2 U_2 \times_3 U_3$. Therefore, the parameter complexity of the proposed ToRA method is $pqv + dp + kq + Tv \sim \mathcal{O}(dp + kq)$ since $T, p, q, v \ll \min(d, k)$. This implies that the parameter complexity of LoRA increases linearly with the number of tasks $T$, while the proposed ToRA method exhibits a sublinear complexity, thereby demonstrating the parameter efficiency of the proposed ToRA method.

## 3.4 TRAINING OBJECTIVE

Under the multi-task learning setting, the training objective function of MTSAM is defined as

$$\mathcal{L}_{MTL} = \frac{1}{\sum w_i} \sum_{i=1}^{T} w_i \mathcal{L}_i, \quad \text{where} \quad \mathcal{L}_i = \frac{1}{n_i} \sum_{j=1}^{n_i} \ell_i(y_i^j, f(x_i^j)). \tag{8}$$

In this formulation, $\mathcal{L}_i$ represents the loss for task $i$ and $w_i$ denotes the corresponding loss weight, $x_i^j$ is the $j$-th training sample for task $i$, $y_i^j$ is the ground truth label, $f(\cdot)$ is the MTSAM model, and $\ell_i$ denotes the loss function specific to task $i$.

According to the High-Order Singular Value Decomposition (HOSVD) (De Lathauwer et al., 2000) used in the analysis shown in the next section, every high-order tensor can be decomposed into a core tensor $\mathcal{G}$ and orthogonal (or unitary) matrices. Inspired by this, we utilize the orthogonal

regularization to enforce the orthogonality of $U_1$, $U_2$ and core tensor $\mathcal{G}$ through the final dimension to reduce the redundancy. Consequently, this can be formulated as a regularization term:

$$R(U_1, U_2, \mathcal{G}) = \|U_1^T U_1 - I\|_F^2 + \|U_2^T U_2 - I\|_F^2 + \sum_{l=1}^{v} \|\mathcal{G}(:,:,l)^T \mathcal{G}(:,:,l) - I\|_F^2, \quad (9)$$

where $\|\cdot\|_F$ denotes the Frobenius norm for matrices and $I$ denotes the identity matrix with an appropriate size. Therefore, the overall objective function of the MTSAM is formulated as

$$\mathcal{L}_{total} = \mathcal{L}_{MTL} + \lambda R(U_1, U_2, \mathcal{G}), \quad (10)$$

where $\lambda$ is the hyper-parameter that controls the impact of orthogonal regularization.

## 3.5 ANALYSIS

In this section, we analyze the advantages of ToRA over LoRA in terms of the expressive power. For LoRA, Zeng & Lee (2024) apply the best low-rank approximation (Eckart & Young, 1936) of the error matrix under the spectral norm. For the parameter matrix in a single layer, let $\overline{W}_t$ denote the target parameter matrix for task $t$, and define the error matrix as $E_t = \overline{W}_t - W$. When using LoRA, the update parameter is denoted by $\Delta W_t$, and the minimum difference between the adapted and target models is given by

$$\min_{\Delta W_t} \left\| (W + \Delta W_t) - \overline{W}_t \right\|_2 = \sigma_{r+1}(E_t), \quad (11)$$

where $\sigma_r(E_t)$ denotes the $r$-th largest singular value of $E_t$. For ToRA, the optimal approximation is formulated as

$$\begin{aligned} \min \quad & \|\mathbf{E} - \mathcal{G} \times_1 U_1 \times_2 U_2 \times_3 U_3\|_F^2 \\ \text{s.t.} \quad & U_k^\top U_k = I, \ \mathrm{rank}(U_k) \le r_k, \end{aligned} \quad (12)$$

where the objective is inherently complex, and the best approximation may not always exist (Kolda & Bader, 2009). Therefore, measuring the expressive power of ToRA can be challenging. However, as proven in Theorem 1 below, we demonstrate that for any multi-task learning problem solvable by multiple LoRAs, ToRA can also solve the problem by using fewer parameters. This implies that ToRA has superior expressive power for multi-task learning when compared with multiple LoRAs.

**Theorem 1.** (ToRA's Superiority over Multiple LoRAs in Expressive Power) *Assume there are $T$ LoRAs, whose update parameter matrix is denoted by $\Delta W_t$, designed to solve each task $t$ with a rank $r_t$. Let $\Delta \mathbf{W}$ represent the update parameter tensor and $\Delta \mathbf{W}_{(1)}$, $\Delta \mathbf{W}_{(2)}$ denote the flattened tensor corresponding to the vertical and horizontal concatenation of $\{\Delta W_t\}_{t=1}^T$. Define $p$ and $q$ as the rank of $\Delta \mathbf{W}_{(1)}$ and $\Delta \mathbf{W}_{(2)}$, respectively. Then, there exists a ToRA with core tensor $\mathcal{G} \in \mathbb{R}^{p \times q \times T}$, and factor matrices $U_1 \in \mathbb{R}^{d \times p}$, $U_2 \in \mathbb{R}^{k \times q}$, $U_3 \in \mathbb{R}^{T \times T}$ such that the Tucker decomposition $\mathcal{G} \times_1 U_1 \times_2 U_2 \times_3 U_3$ reconstructs $\Delta \mathbf{W}$. Furthermore, ToRA utilizes fewer parameters, satisfying $(dp + kq) \le \sum_t (d + k) r_t$.*

## 4 EXPERIMENTS

In this section, we empirically evaluate the proposed MTSAM on three benchmark datasets, including *NYUv2* (Silberman et al., 2012), *CityScapes* (Cordts et al., 2016), and *PASCAL-Context* (Everingham et al., 2010).

**Baselines.** We compare the proposed MTSAM with several baselines, including CNN-based methods (i.e., Single-Task Learning (STL), Hard-Parameter Sharing (HPS), Cross-Stitch (Misra et al., 2016), Multi-Task Attention Network (MTAN) (Liu et al., 2019), and NDDR-CNN (Gao et al., 2019)), Transformer-based approaches (i.e., VTAGML (Bhattacharjee et al., 2023) and Swin-MTL (Taghavi et al., 2024)), and the method using cross-attention (i.e. DenseMTL (Lopes et al., 2023)). To evaluate the effectiveness of the proposed ToRA method, we also compare with LoRA-STL, LoRA-HPS, and MultiLoRA (Wang et al., 2023) that all fine-tune the MTSAM.

**Evaluation metric.** For the three datasets, we use multiple metrics to evaluate the performance on each task and we put the introduction of them in Appendix B.1. Moreover, following the setup

Table 1: Performance on three tasks (i.e., 13-class semantic segmentation, depth estimation, and surface normal prediction) of the *NYUv2* dataset. The best results for each task are shown in **bold**. ↑(↓) means that the higher (lower) the value, the better the performance. The number of trainable parameters (i.e., Params.) is calculated in MB.

| Method | Segmentation | | Depth | | Surface Normal | | | | | Param. (M)↓ | $\Delta_b$ ↑ |
| | | | | | Angle Distance | | Within $t°$ | | | | |
| | mIoU↑ | Pix Acc↑ | Abs Err ↓ | Rel Err↓ | Mean ↓ | Median ↓ | 11.25 ↑ | 22.5 ↑ | 30 ↑ | | |
|---|---|---|---|---|---|---|---|---|---|---|---|
| HPS | 54.48 | 75.82 | 0.3839 | 0.1548 | 23.50 | 17.06 | 35.31 | 61.10 | 72.14 | 71.89 | +0.00% |
| STL | 53.98 | 75.38 | 0.3945 | 0.1631 | 22.25 | 15.63 | 38.12 | 64.38 | 74.81 | 118.91 | +0.45% |
| Cross-Stitch | 53.46 | 75.49 | 0.3804 | 0.1555 | 23.01 | 16.33 | 37.01 | 62.42 | 73.02 | 118.89 | +0.66% |
| MTAN | 54.74 | 75.78 | 0.3796 | 0.1549 | 22.97 | 16.30 | 36.91 | 62.63 | 73.32 | 92.35 | +0.77% |
| NDDR-CNN | 53.84 | 75.23 | 0.3871 | 0.1560 | 22.60 | 16.07 | 37.67 | 63.43 | 73.92 | 169.10 | +0.91% |
| VTAGML | 58.60 | 78.63 | 0.3716 | 0.1525 | 22.05 | 15.70 | 38.14 | 64.28 | 74.50 | 314.15 | +4.70% |
| DenseMTL | 56.65 | 77.68 | 0.3569 | 0.1391 | 22.03 | 15.87 | 37.25 | 64.67 | 75.47 | 423.45 | +5.88% |
| SwinMTL | 64.23 | 82.78 | **0.2841** | **0.1129** | 18.94 | 13.34 | 43.35 | 71.32 | 80.89 | 333.91 | +19.55% |
| LoRA-HPS (r=32) | 56.77 | 78.37 | 0.3470 | 0.1412 | 18.97 | 13.41 | 44.56 | 71.20 | 80.68 | 58.84 | +10.67% |
| LoRA-STL (r=16) | 62.06 | 81.72 | 0.3124 | 0.1233 | 16.44 | 11.39 | 51.04 | 77.01 | 85.31 | 64.83 | +20.25% |
| LoRA-STL (r=32) | 58.34 | 78.61 | 0.3330 | 0.1335 | 16.54 | 11.42 | 51.08 | 76.65 | 84.99 | 82.84 | +16.34% |
| MultiLoRA | 64.85 | 83.07 | 0.3113 | 0.1220 | 17.26 | 12.19 | 48.28 | 74.65 | 83.58 | 65.12 | +20.11% |
| MTSAM | **65.98** | **83.42** | 0.2898 | 0.1140 | **16.34** | **11.33** | **51.22** | **77.20** | **85.51** | 59.59 | **+23.93%** |

in (Maninis et al., 2019), we use the average of the relative improvement of each task over the HPS architecture as another evaluation metric, which is formulated as

$$\Delta_b = \frac{1}{T} \sum_{i=1}^{T} \frac{1}{K_i} \sum_{j=1}^{K_i} \frac{(-1)^{s_{i,j}}(M_{i,j}^b - M_{i,j}^{HPS})}{M_{i,j}^{HPS}},$$

where $T$ denotes the number of tasks, $K_i$ denotes the number of metrics for task $i$, $M_{i,j}^b$ and $M_{i,j}^{HPS}$ denote the performance of the method $b$ and the HPS architecture for the $j$th metric in task $i$, respectively, and $s_{i,j}$ is set to 1 if a lower value indicates better performance in terms of the $j$th metric in task $i$ and otherwise 0.

**Implementation details.** The batch size is set to 4 for *NYUv2* and 8 for *CityScapes* and *PASCAL-Context*. The cross-entropy loss, $L_1$ loss, and cosine similarity loss are used as the loss functions of the semantic segmentation, depth estimation, and surface normal prediction tasks, respectively. The Adam optimizer is used to update fine-tuned parameters. In the Adam optimizer, an initial learning rate is set to $10^{-3}$, the linear learning rate scheduler with warmup is adopted while the warmup rate is set to 0.05, and the weight decay is set to $10^{-6}$. The dropout rate is set to 0.1. For the proposed ToRA, we set $p = q = 32$, $v = 8$ on the *NYUv2* and *PASCAL-Context* datasets, and $p = q = 16$, $v = 4$ on the *CityScapes* dataset. The hyper-parameter $\lambda$ is set to 1. The total number of fine-tuned epochs is set to 200, 50, and 30 for the *NYUv2*, *CityScapes*, and *PASCAL-Context* datasets, respectively. For *CityScapes* and *PASCAL-Context* datasets, we use equal weights for each task (i.e., $w_i$ equals 1 in Eq. (8)), while for the *NYUv2* dataset, we follow (Lopes et al., 2023) to set the weights of semantic segmentation, depth estimation, and surface normal prediction tasks to be 1, 1, and 4, respectively.

## 4.1 RESULTS

The results on the *NYUv2*, *CityScapes*, and *PASCAL-Context* datasets are presented in Tables 1, 2, and 3, respectively. As can be seen, MTSAM achieves the best average performance across all datasets in terms of $\Delta_b$ compared to all baselines. Furthermore, MTSAM demonstrates better parameter efficiency, offering advantages in storage and enhancing its practical application value. Moreover, LoRA-STL, which employs separate LoRAs for each task, possesses better performance than LoRA-HPS with a shared LoRA. This demonstrates the importance of utilizing task-specific components. The superior performance of MTSAM with ToRA over both LoRA-STL and LoRA-HPS suggests that ToRA effectively leverages both task-shared and task-specific information, thereby improving overall performance.

## 4.2 ABLATION STUDY

**Sensitivity w.r.t. rank.** We conduct sensitivity analysis to evaluate the impact of ranks of ToRA (i.e., $p$, $q$, and $v$) to the performance of MTSAM while keeping the remaining hyper-parameters

Table 2: Performance on two tasks (i.e., 7-class semantic segmentation and depth estimation) in the *CityScapes* dataset. The best results for each task are shown in **bold**. ↑(↓) means that the higher (lower) the value, the better the performance. The number of trainable parameters (i.e., Params.) is calculated in MB.

| Method | Segmentation | | Depth | | Param. (M)↓ | $\Delta_b$ ↑ |
|---|---|---|---|---|---|---|
| | mIoU↑ | Pix Acc ↑ | Abs Err ↓ | Rel Err↓ | | |
| HPS | 67.40 | 90.92 | 0.0142 | 45.4262 | 55.76 | +0.00% |
| STL | 68.13 | 91.28 | 0.0133 | 45.0390 | 79.27 | +2.17% |
| Cross-Stitch | 68.01 | 91.29 | 0.0135 | 44.4246 | 79.27 | +2.11% |
| MTAN | 68.97 | 91.59 | 0.0136 | 43.7508 | 72.04 | +2.74% |
| NDDR-CNN | 68.02 | 91.25 | 0.0137 | 44.8662 | 101.58 | +1.51% |
| VTAGML | 73.70 | 93.23 | 0.0138 | 42.8304 | 288.35 | +5.10% |
| DenseMTL | 69.75 | 91.45 | 0.0152 | 52.1401 | 333.40 | −4.44% |
| SwinMTL | 73.33 | 92.87 | 0.0132 | 38.0720 | 333.31 | +8.54% |
| LoRA-HPS (r=16) | 86.23 | 96.30 | 0.0123 | 34.2000 | 37.35 | +17.98% |
| LoRA-STL (r=8) | 85.86 | 96.26 | 0.0110 | 34.3000 | 37.35 | +20.07% |
| LoRA-STL (r=16) | 82.64 | 95.28 | **0.0107** | 33.4312 | 43.35 | +19.62% |
| MultiLoRA | 87.23 | 96.67 | 0.0127 | 31.0091 | 46.81 | +19.51% |
| MTSAM | **87.45** | **96.80** | 0.0113 | **33.0086** | 37.44 | **+20.99%** |

Table 3: Performance on four tasks (i.e., 21-class semantic segmentation, 7-class human parts segmentation, saliency estimation, and surface normal estimation) in the *PASCAL-Context* dataset. The best results for each task are shown in **bold**. ↑(↓) means that the higher (lower) the value, the better the performance. The number of trainable parameters (i.e., Params.) is calculated in MB.

| Method | Seg.↑ | H.Parts↑ | Sal.↑ | Normal↓ | Param. (M) | $\Delta_b$ ↑ |
|---|---|---|---|---|---|---|
| HPS | 64.77 | 57.91 | 64.10 | **14.21** | 30.07 | +0.00% |
| STL | 65.14 | 58.58 | 65.02 | 15.94 | 63.60 | −2.25% |
| Cross-Stitch | 64.97 | 58.63 | 64.46 | 15.32 | 79.46 | −1.42% |
| MTAN | 64.56 | 59.08 | 64.57 | 14.74 | 36.61 | −0.33% |
| NDDR-CNN | 65.28 | 59.18 | 65.09 | 15.57 | 69.25 | −1.26% |
| Hyperformer | 71.43 | 60.73 | 65.54 | 17.77 | 287.32 | −1.91% |
| Polyhistor | 70.87 | 59.54 | 65.47 | 17.47 | 34.18 | −2.14% |
| Polyhistor-Lite | 70.24 | 59.12 | 64.75 | 17.40 | 11.29 | −2.72% |
| LoRA-HPS (r=32) | 48.19 | 46.73 | 69.50 | 20.38 | 74.33 | −19.97% |
| LoRA-STL (r=16) | 55.25 | 71.33 | 75.72 | 17.05 | 86.33 | +1.65% |
| LoRA-STL (r=32) | 65.07 | **72.05** | **76.41** | 16.82 | 110.33 | +6.42% |
| MultiLoRA | 72.39 | 67.78 | 71.66 | 20.07 | 92.80 | −0.16% |
| MTSAM | **74.13** | 71.04 | 76.28 | 17.10 | 74.71 | **+8.95%** |

Table 4: Ablation studies on the impact of rank on the *NYUv2* dataset. The best results for each task are shown in **bold**. ↑(↓) means that the higher (lower) the value, the better the performance. The number of trainable parameters (i.e., Params.) is calculated in MB.

| Method | Segmentation | | Depth | | Surface Normal | | | | | Param. (M)↓ | $\Delta_b$ ↑ |
|---|---|---|---|---|---|---|---|---|---|---|---|
| | | | | | Angle Distance | | Within $t°$ | | | | |
| | mIoU↑ | Pix Acc ↑ | Abs Err ↓ | Rel Err↓ | Mean ↓ | Median ↓ | 11.25 ↑ | 22.5 ↑ | 30 ↑ | | |
| MTSAM ($p = q = 16, v = 8$) | 64.66 | 83.15 | 0.2966 | 0.1153 | 16.40 | 11.42 | 50.89 | 76.85 | 85.25 | 53.03 | +22.85% |
| MTSAM ($p = q = 32, v = 4$) | 65.57 | 83.29 | **0.2888** | 0.1149 | **16.29** | **11.32** | **51.22** | **77.40** | **85.64** | 59.21 | +23.77% |
| MTSAM ($p = q = 32, v = 8$) | **65.98** | **83.42** | 0.2898 | **0.1140** | 16.34 | 11.33 | **51.22** | 77.20 | 85.51 | 59.59 | +23.93% |

consistent with previous experiments. According to the results shown in Table 4, we can see that the proposed ToRA method consistently outperforms the LoRA-STL and LoRA-HPS methods across different combinations of ranks, demonstrating the effectiveness of the proposed ToRA method.

**Impact of orthogonal regularization.** We conduct ablation studies on *NYUv2* dataset to evaluate the impact of the orthogonal regularization on $U_1$, $U_2$ and the core tensor $\mathcal{G}$ by employing the same hyper-parameter settings as previous experiments. We compare MTSAM with that without the orthogonal regularization on $\mathcal{G}$ (denoted by MTSAM (w/o $\mathcal{G}$)) and without the orthogonal regularization on $U_1$, $U_2$ and $\mathcal{G}$ (denoted by MTSAM (w/o $U_1, U_2, \mathcal{G}$)). According to the results shown in Table 5, we can see that the orthogonality regularization on $\mathcal{G}$ effectively improves the performance across various tasks, demonstrating the effectiveness of the orthogonal regularization.

Table 5: Ablation studies on the impact of orthogonal regularization on the *NYUv2* dataset. Best results are shown in **bold**. ↑(↓) means that the higher (lower) the value, the better the performance.

| Method | Segmentation | | Depth | | Surface Normal | | | | | $\Delta_b$ ↑ |
|---|---|---|---|---|---|---|---|---|---|---|
| | | | | | Angle Distance | | Within $t°$ | | | |
| | mIoU↑ | Pix Acc ↑ | Abs Err ↓ | Rel Err↓ | Mean ↓ | Median ↓ | 11.25 ↑ | 22.5 ↑ | 30 ↑ | |
| MTSAM (w/o $U_1, U_2, \mathcal{G}$) | 58.71 | 80.01 | 0.3309 | 0.1305 | 16.88 | 11.66 | 50.17 | 76.05 | 84.48 | +17.01% |
| MTSAM (w/o $\mathcal{G}$) | 65.22 | 83.11 | 0.2984 | 0.1195 | 16.56 | 11.53 | 50.59 | 76.63 | 85.03 | +22.30% |
| MTSAM | **65.98** | **83.42** | **0.2898** | **0.1140** | **16.34** | **11.33** | **51.22** | **77.20** | **85.51** | **+23.93%** |

Table 6: The performance of MTSAM with varying $\lambda$ on the *NYUv2* dataset. The best results for each task are shown in **bold**. ↑(↓) means that the higher (lower) the value, the better the performance. The number of trainable parameters (i.e., Params.) is calculated in MB.

| Method | Segmentation | | Depth | | Surface Normal | | | | | $\Delta_b$ ↑ |
|---|---|---|---|---|---|---|---|---|---|---|
| | | | | | Angle Distance | | Within $t°$ | | | |
| | mIoU↑ | Pix Acc ↑ | Abs Err ↓ | Rel Err↓ | Mean ↓ | Median ↓ | 11.25 ↑ | 22.5 ↑ | 30 ↑ | |
| $\lambda = 0.5$ | 66.05 | **83.82** | 0.2898 | 0.1137 | 16.70 | 11.73 | 49.81 | 76.39 | 84.92 | +23.410% |
| $\lambda = 1.0$ | 65.98 | 83.42 | 0.2898 | 0.1140 | **16.34** | **11.33** | **51.22** | **77.20** | **85.51** | +23.931% |
| $\lambda = 1.5$ | **66.11** | 83.50 | **0.2872** | **0.1135** | 16.47 | 11.48 | 50.68 | 77.05 | 85.39 | +23.933% |

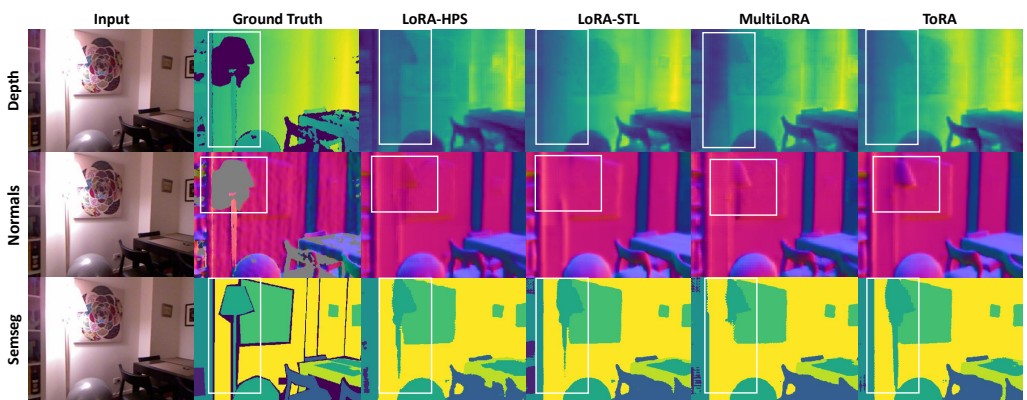

Figure 5: Comparison among predictions of LoRA-HPS, LoRA-STL, MultiLoRA, and ToRA to fine-tune MTSAM on the *NYUv2* dataset.

**Sensitivity w.r.t. hyper-parameter** $\lambda$**.** We explore the sensitivity of the performance of ToRA with respect to hyper-parameter $\lambda$. According to the results shown in Table 6, we can see that MTSAM is not so sensitive to the hyper-parameter $\lambda$ over $[0.5, 1.5]$, making the setting of $\lambda$ not so difficult.

### 4.3 QUALITATIVE EVALUATION

Figure 5 shows the predictions of the MTSAM fine-tuned with LoRA-STL, LoRA-HPS, Multi-LoRA, and ToRA on the *NYUv2* dataset, respectively. More qualitative results are shown in Figures 6-11 in Appendix D. As can be seen, the prediction results of ToRA are better than the baselines for different tasks. As shown in the white boxes, the proposed ToRA method generates more accurate results than the baseline methods given the ground truth when dealing with vague and slender objects. Therefore, the proposed MTSAM fine-tuned with ToRA achieves the best performance in both qualitative and quantitative evaluations.

## 5 CONCLUSION

In this paper, we propose the MTSAM, which modifies the architecture of SAM and leverages a low-rank tensor decomposition method to fine-tune the encoder of MTSAM. MTSAM introduces task embeddings to generate outputs with the corresponding number of channels, enabling the model can be adapted to different tasks. The proposed ToRA can use both task-shared and task-specific information during the multi-task fine-tuning process. The experimental results demonstrate the effectiveness of MTSAM. In future work, we are interested in applying MTSAM to more applications.

ACKNOWLEDGEMENTS

This work is supported by National Key R&D Program of China 2022ZD0160300 and NSFC key grant under grant no. 62136005.

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

# A    PROOFS

In this section, we present the proof of Thereom 1, demonstrating the superiority of ToRA over multiple LoRAs. We begin by introducing higher-order singular value decomposition (HOSVD) proposed by De Lathauwer et al. (2000) as our Lemma 1.

**Lemma 1.** (HOSVD) *Every tensor* $\mathbf{A} \in \mathbb{R}^{d_1 \times d_2 \times d_3}$ *can be decomposed into a core tensor* $\mathcal{G} \in \mathbb{R}^{d_1 \times d_2 \times d_3}$ *and the left singular vectors* $U_k \in \mathbb{R}^{d_k \times d_k}$ *of* $k$*-mode flattened matrix* $\mathbf{A}_{(k)}$*, as follows*

$$
\begin{aligned}
\mathbf{A} &= \mathbf{A} \times_1 (U_1 U_1^\top) \times_2 (U_2 U_2^\top) \times_3 (U_3 U_3^\top) \\
&= \left( \mathbf{A} \times_1 U_1^\top \times_2 U_2^\top \times_3 U_3^\top \right) \times_1 U_1 \times_2 U_2 \times_3 U_3 \\
&= \mathcal{G} \times_1 U_1 \times_2 U_2 \times_3 U_3.
\end{aligned}
$$

*According to* Property 10 *of (De Lathauwer et al., 2000), we can achieve a low-rank approximation of tensor through compact HOSVD. Let the rank of* $\mathbf{A}_{(k)}$ *be* $R_k$*, and define a tensor* $\hat{\mathbf{A}}$ *by discarding the smallest singular values* $\sigma_{I_k'+1}^{(k)}, \sigma_{I_k'+2}^{(k)}, \ldots, \sigma_{R_k}^{(k)}$ *for given values of* $I_k'$*, that is, we set the corresponding parts of the core tensor* $\mathcal{G}$ *to zero. Then we can get*

$$
\|\mathcal{A} - \hat{\mathcal{A}}\|^2 \leq \sum_{i_1 = I_1'+1}^{R_1} \sigma_{i_1}^{(1)^2} + \sum_{i_2 = I_2'+1}^{R_2} \sigma_{i_2}^{(2)^2} + \cdots + \sum_{i_N = I_N'+1}^{R_N} \sigma_{i_N}^{(N)^2}.
$$

**Lemma 2.** *Given* $K$ *matrices* $A_k \in \mathbb{R}^{m \times n}$*, let Rank denote the rank of a matrix. It follows trivially that* $Rank([A_1 \; A_2 \; \ldots \; A_K]) \leq \sum_k Rank(A_k)$ *and* $Rank([A_1^\top \; A_2^\top \; \ldots \; A_K^\top]) \leq \sum_k Rank(A_k)$*.*

**Theorem 1.** (ToRA's Superiority over Multiple LoRAs in Expressive Power) *Assume there are* $T$ *LoRAs, whose update parameter matrix is denoted by* $\Delta W_t$*, designed to solve each task* $t$ *with a rank* $r_t$*. Let* $\Delta \mathbf{W}$ *represent the update parameter tensor and* $\Delta \mathbf{W}_{(1)}$*,* $\Delta \mathbf{W}_{(2)}$ *denote the flattened tensor corresponding to the vertical and horizontal concatenation of* $\{\Delta W_t\}_{t=1}^T$*. Define* $p$ *and* $q$ *as the rank of* $\Delta \mathbf{W}_{(1)}$ *and* $\Delta \mathbf{W}_{(2)}$*, respectively. Then, there exists a ToRA with core tensor* $\mathcal{G} \in \mathbb{R}^{p \times q \times T}$*, and factor matrices* $U_1 \in \mathbb{R}^{d \times p}$*,* $U_2 \in \mathbb{R}^{k \times q}$*,* $U_3 \in \mathbb{R}^{T \times T}$ *such that the Tucker decomposition* $\mathcal{G} \times_1 U_1 \times_2 U_2 \times_3 U_3$ *reconstructs* $\Delta \mathbf{W}$*. Furthermore, ToRA utilizes fewer parameters, satisfying* $(dp + kq) \leq \sum_t (d + k) r_t$*.*

*Proof.* The weight updates $\Delta \mathbf{W}$ of multiple LoRAs across $T$ tasks can be organized into two flattened forms: $\Delta \mathbf{W}_{(1)}$ and $\Delta \mathbf{W}_{(2)}$. Specially, we define these as follows

$$
\begin{aligned}
\Delta \mathbf{W}_{(1)} &= [\Delta W_1 \; \Delta W_2 \; \ldots \; \Delta W_T], \\
\Delta \mathbf{W}_{(2)}^\top &= [\Delta W_1^\top \; \Delta W_2^\top \; \ldots \; \Delta W_T^\top].
\end{aligned}
$$

According to Lemma 1, we can express $\Delta \mathbf{W}$ using a core tensor $\mathcal{G} \in \mathbb{R}^{p \times q \times T}$ with three unitary matrices $U_1 \in \mathbb{R}^{d \times p}$, $U_2 \in \mathbb{R}^{k \times q}$, $U_3 \in \mathbb{R}^{T \times T}$. These unitary matrices represent the left singular vectors corresponding to the three flatten matrices of $\Delta \mathbf{W}$ and $\mathcal{G} = \Delta \mathbf{W} \times_1 U_1^\top \times_2 U_2^\top \times_3 U_3^\top$.

From Lemma 2, we have the inequalities $p \leq \sum_t r_t$ and $q \leq \sum_t r_t$. Consequently, we deduce that $dp + kq \leq \sum_t (d + k) r_t$. Given that $T, p, q, v$ are significantly smaller than $\min(d, k)$, we conclude that ToRA can achieve the same weight updates with fewer parameters. In particular, this theorem constructs a specific instance, indicating that the expressive power of ToRA will be greater. $\qquad \square$

# B    EXPERIMENTAL DETAILS

## B.1    METRIC FOR EACH TASK

**NYUv2 and CityScapes datasets.** For the semantic segmentation task, we use the mean Intersection over Union (mIoU) and Pixel Accuracy (Pix Acc) to evaluate. For the depth prediction task, we use the Absolute Error (Abs Err) and Real Error (Rel Err) to evaluate. For the surface normal estimation task, we use the mean and the median of angular error measured in degrees and the percentage of pixels whose angular error is within 11.25, 22.5, and 30 degrees to evaluate.

Table 7: Performance on three tasks (i.e., 13-class semantic segmentation, depth estimation, and surface normal prediction) of the *NYUv2* dataset. The best results for each task are shown in **bold**. ↑(↓) means that the higher (lower) the value, the better the performance. The number of trainable parameters (i.e., Params.) is calculated in MB.

| Method | Segmentation | | Depth | | Surface Normal | | | | | Param. (M)↓ | $\Delta_b$ ↑ |
| | | | | | Angle Distance | | Within $t°$ | | | | |
| | mIoU↑ | Pix Acc↑ | Abs Err ↓ | Rel Err↓ | Mean ↓ | Median ↓ | 11.25 ↑ | 22.5 ↑ | 30 ↑ | | |
| Full fine-tuning | 58.76 | 80.18 | 0.3063 | 0.1223 | 19.61 | 13.76 | 43.14 | 69.44 | 79.17 | 1222.47 | +14.57% |
| MultiLoRA | 64.85 | 83.07 | 0.3113 | 0.1220 | 17.26 | 12.19 | 48.28 | 74.65 | 83.58 | 65.12 | +20.11% |
| Terra | 60.98 | 80.93 | 0.3461 | 0.1343 | 18.47 | 13.27 | 44.89 | 71.84 | 81.48 | 52.86 | +13.70% |
| HydraLoRA | 66.46 | 83.90 | 0.3033 | 0.1202 | 16.86 | 11.60 | 49.47 | 75.86 | 84.54 | 71.30 | +22.11% |
| MTSAM | **65.98** | **83.42** | 0.2898 | 0.1140 | **16.34** | **11.33** | **51.22** | **77.20** | **85.51** | 59.59 | **+23.93%** |

**PASCAL-Context dataset.** For the semantic segmentation task, human parts segmentation task, and saliency estimation task, we use the mean Intersection over Union (mIoU) to evaluate. For the surface normal estimation task, we use the mean of angular error measured in degrees to evaluate.

## C  MORE EXPERIMENT

### C.1  RESULTS

To evaluate the effectiveness of the proposed ToRA method, we also compare with LoRA-based methods (i.e., MultiLoRA (Wang et al., 2023), Terra (Zhuang et al., 2024), and HydraLoRA (Tian et al., 2024)) and the full fine-tuning method that fine-tunes the entire MTSAM. The results are shown in Table 7. As can be seen, ToRA achieves better performance and parameter efficiency compared to LoRA-based methods and full fine-tuning.

### C.2  ABLATION STUDY

Table 8: The performance of MTSAM with modified MLP layer and task embeddings on *NYUv2* dataset.

| Method | Segmentation | | Depth | | Surface Normal | | | | | Param. (M)↓ | $\Delta_b$ ↑ |
| | | | | | Angle Distance | | Within $t°$ | | | | |
| | mIoU↑ | Pix Acc ↑ | Abs Err ↓ | Rel Err↓ | Mean ↓ | Median ↓ | 11.25 ↑ | 22.5 ↑ | 30 ↑ | | |
| MLP | 63.50 | 82.45 | 0.3294 | 0.1235 | 17.93 | 13.02 | 45.46 | 73.28 | 82.86 | 65.66 | +17.35% |
| Task embeddings | **65.98** | **83.42** | **0.2898** | **0.1140** | **16.34** | **11.33** | **51.22** | **77.20** | **85.51** | 59.59 | **+23.93%** |

**Impact of task embeddings.** To demonstrate the effectiveness of the proposed task embedding, we compared it with the method of modifying the MLP output dimensions for different tasks on the *NYUv2* dataset. As shown in Table 8, task embedding performs better. This improvement is due to the interaction between task embeddings and image features through the cross-attention mechanism, which enables the decoder to better learn the task-specific knowledge and achieve superior results.

## D  MORE QUALITATIVE EVALUATIONS

Figures 6, 7, 8, 9 and 10 show the predictions of the MTSAM fine-tuned with LoRA-STL, LoRA-HPS, MultiLoRA, and ToRA on the *NYUv2* and *CityScapes* datasets, respectively. Figure 11 shows the prediction of MTSAM on high-resolutional images. It can be observed that the predictions of MTSAM outperform those of other baselines in different tasks and datasets. In the areas highlighted by the white boxes, MTSAM generates more accurate results. Therefore, using MTSAM with ToRA yields better performance.

## E  QUALITATIVE EVALUATIONS ON ZERO-SHOT ABILITY

To evaluate MTSAM's performance on unseen data, we applied the model fine-tuned on the *NYUv2* dataset to make depth predictions on *CityScapes* dataset. Qualitative results are shown in Figure 12,

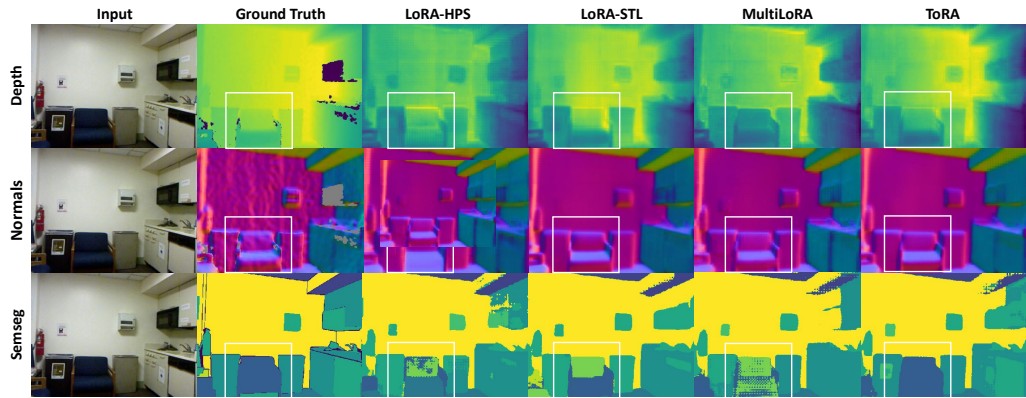

Figure 6: (1/3) Comparison among predictions of LoRA-HPS, LoRA-STL, MultiLoRA, and ToRA to fine-tune MTSAM on the *NYUv2* dataset.

and illustrate that MTSAM is capable of handling unseen data distributions to some extent. However, it is important to note that the *NYUv2* dataset consists of indoor images, whereas the *CityScapes* dataset comprises of outdoor images, leading to significant differences in depth distribution and object types. Additionally, the two datasets differ in terms of resolution and the hardware used for groud-truth depth predictions. Consequently, MTSAM exhibits some inaccuracies, particularly for distant objects.

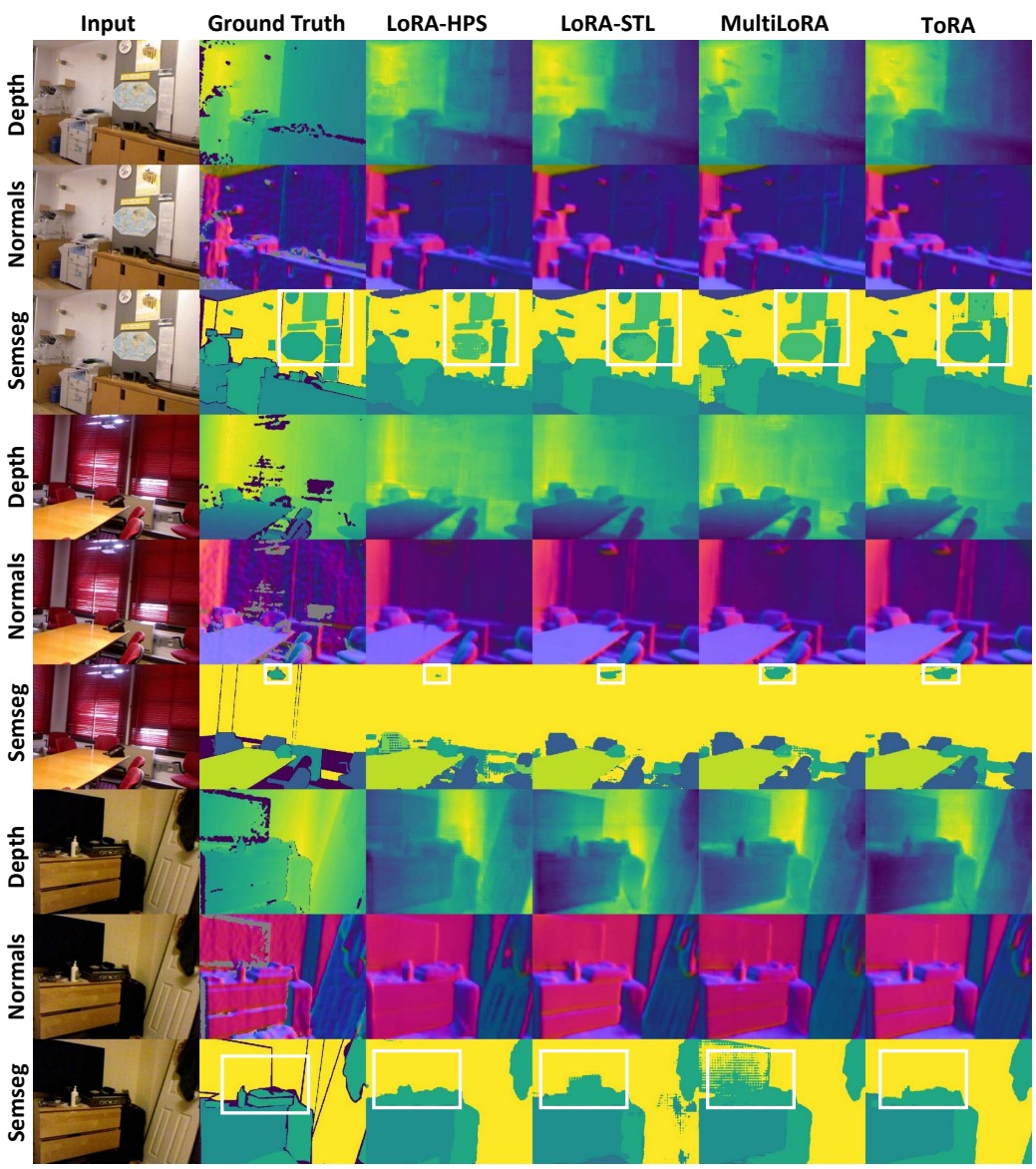

Figure 7: (2/3) Comparison among predictions of LoRA-HPS, LoRA-STL, MultiLoRA, and ToRA to fine-tune MTSAM on the *NYUv2* dataset.

Figure 8: (3/3) Comparison among predictions of LoRA-HPS, LoRA-STL, MultiLoRA, and ToRA to fine-tune MTSAM on the *NYUv2* dataset.

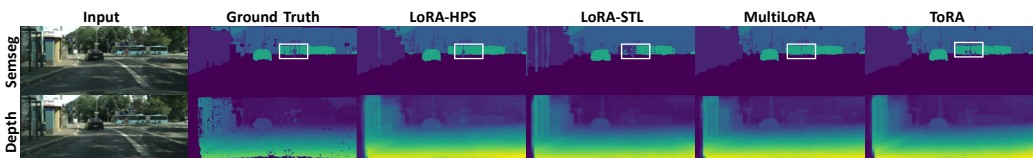

Figure 9: (1/2) Comparison among predictions of LoRA-HPS, LoRA-STL, MultiLoRA, and ToRA to fine-tune MTSAM on the *CityScapes* dataset.

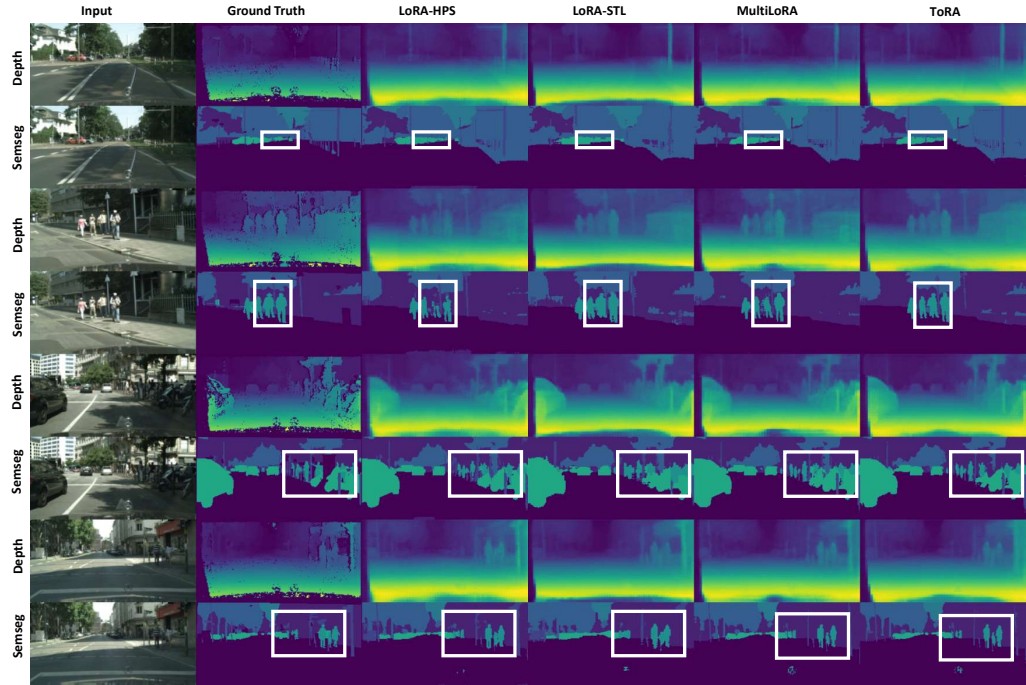

Figure 10: (2/2) Comparison among predictions of LoRA-HPS, LoRA-STL, MultiLoRA, and ToRA to fine-tune MTSAM on the *CityScapes* dataset.

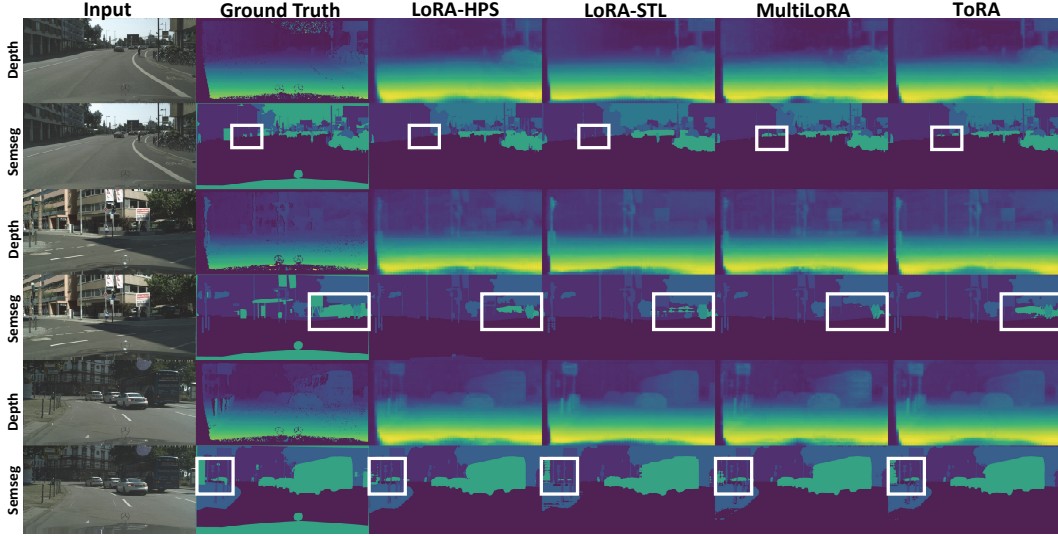

Figure 11: Comparison among predictions of LoRA-HPS, LoRA-STL, MultiLoRA, and ToRA to fine-tune MTSAM on high-quality images.

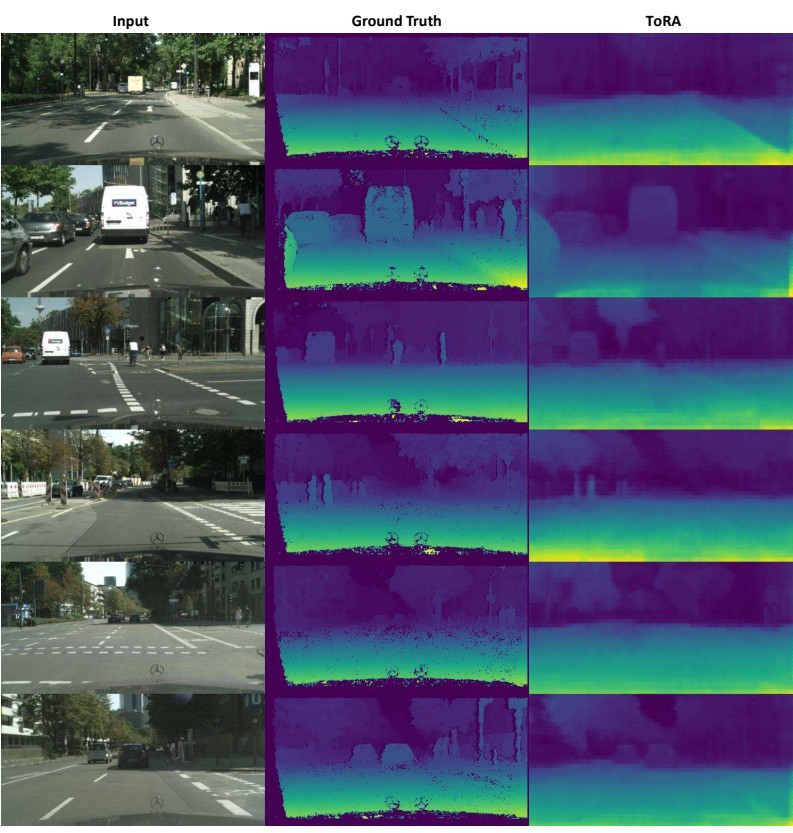

Figure 12: Zero-shot depth estimation of MTSAM which is trained on the *NYUv2* dataset and evaluated on the CityScapes dataset.

