# OpenReview forum: "MTSAM: Multi-Task Fine-Tuning for Segment Anything Model"
_ICLR.cc/2025/Conference — ICLR 2025 Poster_

### Official Review · Reviewer_Zgv2 · 2024-10-31

**Soundness:** 3
**Presentation:** 3
**Contribution:** 4
**Rating:** 6
**Confidence:** 4

**Summary:**

This paper presents an interesting idea to make SAM multi-task capable. As mentioned by the authors (L201-204), despite SAM's success, it's end-to-end adaptability is limited by its prompt-guided paradigm. The paper proposes that one can learn multiple mask decoders, one per task, and then have a TORA (Tensorized Low Rank Adaptation) as opposed to say one LoRA per task to capture the task information. In some sense, the main show is actually TORA, for which the authors' motivation that all the tasks share information while having task-specific requirement makes sense. Experiments are sufficient to support the proposal.

**Strengths:**

- The multi-task challenge with SAM is indeed a real problem that many researchers faced. The original SAM would output multiple potential masks when the prompts are ambiguous which sometimes make it hard for practitioner to adapt it to multiple downstream tasks.
- The main show is TORA which the paper clearly presented why the final formulation is as given. I appreciate this clarity. In fact, I think ToRA can stand as a separate paper exploring its applicability in other problem domains, given that LoRA is a hotly researched topic currently.
- Moreover, ToRA has a lower complexity computational wise. This could be quite valuable when the number of tasks really scale up. I am curious about scaling up the number of tasks.
- The regularization term in Eq 9 is interesting.

**Weaknesses:**

- Scaling-up experiments are lacking. Is there any way to see a larger number of tasks beyond 3-4? I really would like to stress test ToRA.
- Are there any out of domain experiments, same task but out of domain datasets?
- Are there any experiments on the speed? ToRA as described has a computational advantage but there does not seem to be any experiments to back that up, unless I missed them.
- Table 3 is slightly disappointing with LoRA-STL (r-32) beating MTSAM on some tasks. Not a show-stopper since overall performance seems ok.
- The qualitative examples need to be better. Resolution is very low and hard to tell. I know cityscapes are like that, but perhaps testing on some higher res samples would be more convincing.

**Questions:**

See weaknesses.

---

> ### Author Response · Authors · 2024-11-21
> **Response to Reviewer Zgv2 1/2**
>
> We sincerely thank the reviewer for providing valuable comments. You can find our response below for your concerns. Please kindly let us know if you have any further concerns.
>
> > Q1. Scaling-up experiments are lacking. Is there any way to see a larger number of tasks beyond 3-4? I really would like to stress test ToRA.
>
> Thank you for your valuable suggestion. Due to time and computational resource constraints, we conducted few-shot experiments on the *Taskonomy* dataset. Specifically, we used 200 images from six tasks (i.e., segment semantic, depth estimation, surface normal, keypoint detection, edge detection, and reshading) of one view in *Taskonomy* as the training data and used another view for testing. The results are shown in following table. As can be seen, ToRA achieves better performance than LoRA-STL in 5 tasks and has comparable result in 1 task, which demonstrates the effectiveness of the proposed ToRA.
>
> | Method | Param. (M) $\downarrow$ | Seg. $\downarrow$ | Dep. $\downarrow$ | Nor. $\downarrow$ | Key. $\downarrow$ | Edg. $\downarrow$ | Res. $\downarrow$
> | -------- | -------- | -------- | -------- | -------- | -------- | -------- | --------
> LoRA-STL | 165.27 | 0.0143     | 0.9753     | 0.3907     | 0.7317     | 0.3523     | **0.5597**
> ToRA     | 106.03 | **3e-10**  | **0.8550** | **0.2854** | **0.5464** | **0.2561** | 0.5958
>
>
> > Q2. Are there any out of domain experiments, same task but out of domain datasets?
>
> First, we would like to humbly clarify that our proposed framework, MTSAM, is primarily designed for **efficient multi-task fine-tuning of SAM**, rather than for addressing out-of-distribution data scenarios.
>
> Nonetheless, to evaluate its performance on unseen data as suggested, we applied the model fine-tuned on the *NYUv2* dataset to make depth predictions on *CityScapes* dataset. Qualitative results are provided in Appendix E, and illustrate that **MTSAM is capable of handling unseen data distributions** to some extent.
>
>
> > Q3. Are there any experiments on the speed? ToRA as described has a computational advantage but there does not seem to be any experiments to back that up, unless I missed them.
>
> As mentioned in Section 3.3, the proposed method, ToRA, demonstrates better parameter efficiency compared to LoRA, with sublinear growth in parameter w.r.t the number of tasks. Additionally, similar to LoRA, we froze pre-trained parameters and fine-tune low-rank update tensors during training and inference. Consequently, our method has a similar computational complexity to LoRA, resulting in comparable training and inference speeds.

---

> > ### Comment · Reviewer_Zgv2 · 2024-11-25
> >
> > Thanks for your answers.
> >
> > I want to clarify something. When you say it is "multi-task fine-tuning", is it that for a set of new tasks (datasets) you have to fine-tune each time?
> >
> > Sec. 3.3 gave a complexity analysis, and I saw it when I first read the paper. I was asking empirical comparison of the speed, e.g., on the same hardware, same tasks, time taken.
> >
> > Appendix E are qualitative examples, do you have quantitative numbers? It is kind of hard for me to tell, even with the better resolution, which method is better ...

---

> > > ### Author Response · Authors · 2024-11-25
> > > **Response to the Reply of Zgv2**
> > >
> > > Thanks for your timely follow-up questions and for allowing us to provide further clarification.
> > >
> > > > Q1: I want to clarify something. When you say it is "multi-task fine-tuning", is it that for a set of new tasks (datasets) you have to fine-tune each time?
> > >
> > > Yes, you are correct. Our proposed multi-task fine-tuning approach requires fine-tuning for a set of new tasks. Specifically, our aim is to address the challenge of **leveraging an existing general-purpose pre-trained segmentation model, SAM, and fine-tuning it for multiple more specialized downstream tasks**. To address the high cost of fine-tuning separate models for each task, we designed ToRA to use a single adapter capable of solving multiple learning tasks. ToRA achieves this by simultaneously leveraging knowledge from multiple domains, effectively **capturing both task-shared and task-specific information**.
> > >
> > > > Q2: Sec. 3.3 gave a complexity analysis, and I saw it when I first read the paper. I was asking empirical comparison of the speed, e.g., on the same hardware, same tasks, time taken.
> > >
> > > Thank you for your valuable suggestion. We have conducted a comparison in terms of the training speed. The following table reports the training time of one epoch on the Taskonomy dataset under the same setup on the A100 GPU.
> > >
> > > |  | Cost Time (min)|
> > > | --- | --- |
> > > | LoRA-STL | 6.54 |
> > > | MTSAM with ToRA     | 6.67 |
> > >
> > > As you can see, the training speeds of LoRA and ToRA are comparable. It is important to note— as we have emphasized before—that ToRA enables multi-task fine-tuning with fewer parameters, **eliminating the need for separate LoRA modules for each individual task**.
> > >
> > > > Q3: Appendix E are qualitative examples, do you have quantitative numbers? It is kind of hard for me to tell, even with the better resolution, which method is better ...
> > >
> > > Thank you for your valuable suggestion. In Appendix E, we provided qualitative examples to address the concern of "same task but out-of-domain datasets." Specifically, we conducted zero-shot experiments on the depth estimation task by applying models trained on NYUv2 and CityScapes directly to the other dataset. To offer a more comprehensive analysis, we have conducted a **quantitative** evaluation. The results are presented in the following table.
> > >
> > > | Setting  | Abs Err$\downarrow$ |
> > > | -------- | -------- |
> > > | Trained and tested on NYUv2 |  0.2898 |
> > > | Trained on CityScapes, zero-shot tested on NYUv2 | 1.8978 |
> > > | Trained and tested on CityScapes |  0.0113 |
> > > | Trained on NYUv2, zero-shot tested on CityScapes | 0.1197 |
> > >
> > >
> > > As the results indicate, **MTSAM demonstrates zero-shot capability**. From the qualitative examples, we also observe that ToRA performs closely to ground truth in terms of relative depth and object contours (e.g., cars, trees, and people).
> > >
> > > However, **its performance is still not as strong as that of the model trained directly on the corresponding datasets**. This performance gap can be attributed to several factors:
> > > -    There are substantial differences in the data distributions between them, including variations in object categories, depth distribution, lighting conditions, and camera parameters. Specifically, NYUv2 is an indoor dataset, while CityScapes is for outdoor, so cityScapes images often have much larger depth than those in NYUv2, increasing the difficulty of zero-shot learning.
> > > -    Since the decoder of our model is trained from scratch, it may not generalize well to out-of-domain data.

---

> > > > ### Comment · Reviewer_Zgv2 · 2024-11-25
> > > >
> > > > Thanks for the additional results.
> > > >
> > > > I guess what I was asking for is this. I understood it is multi-task FINE TUNING. But I expected that after say fine-tuning on cityscapes for example, it should generalize well to say KITTI, which I believe is in the same domain.
> > > >
> > > > The results for cityscapes and NYU are expected since one is indoor. But I want to know at least it will generalize well to other "task" of the same domain, e.g., cityscapes->KITTI. That, imho, is the gist of this paper.

---

> > > > > ### Author Response · Authors · 2024-11-27
> > > > >
> > > > > Esteemed Reviewer,
> > > > >
> > > > > We are actively working on a response, along with additional experiments, to address your further questions promptly. Since KITTI is a large dataset, we require some time for downloading, uploading, and preprocessing. We appreciate your understanding. We expect to upload the response in 24 hours.
> > > > >
> > > > > Best regards,
> > > > >
> > > > > The Authors

---

> > > > > > ### Author Response · Authors · 2024-11-27
> > > > > >
> > > > > > Thanks for your timely follow-up questions.
> > > > > >
> > > > > > > Q1. I guess what I was asking for is this. I understood it is multi-task FINE TUNING. But I expected that after say fine-tuning on cityscapes for example, it should generalize well to say KITTI, which I believe is in the same domain.
> > > > > >
> > > > > > Thank you for your valuable suggestion. Based on your suggestion, we evaluated MTSAM trained on the CityScapes dataset on the validation set of the Kitti depth estimation task. The results are shown in the table below. As you expected, MTSAM demonstrates strong zero-shot capability on the KITTI dataset.
> > > > > >
> > > > > > | Setting  | Abs Err$\downarrow$ |
> > > > > > | -------- | -------- |
> > > > > > | Trained and tested on CityScapes |  0.0113 |
> > > > > > | Trained on CityScapes, zero-shot tested on KITTI | 0.0607 |

---

> > > > > > > ### Comment · Reviewer_Zgv2 · 2024-11-27
> > > > > > >
> > > > > > > Thanks for the additional work to produce the new result.
> > > > > > >
> > > > > > > Can you explain why you think that MTSAM demonstrates strong zero-shot capability (although I don't think of it zero-shot as cityscapes and kitti are very similar) even thought the error jumps up 5-6 times?

---

> > > > > > > > ### Author Response · Authors · 2024-11-27
> > > > > > > >
> > > > > > > > Thank you for your timely follow-up questions. If you have any further questions or need additional clarification, please do not hesitate to reach out.
> > > > > > > >
> > > > > > > > > Q1. Can you explain why you think that MTSAM demonstrates strong zero-shot capability (although I don't think of it zero-shot as cityscapes and kitti are very similar) even thought the error jumps up 5-6 times?
> > > > > > > >
> > > > > > > > First of all, thank you for the reminder. We will revise the description of "strong zero-shot capability" to simply "zero-shot capability."
> > > > > > > >
> > > > > > > > We think that MTSAM demonstrates zero-shot capability because of its **comparatively better performance** in zero-shot experiments when trained on CityScapes and tested on KITTI (0.0607), as opposed to other scenarios, such as training on NYUv2 and testing on CityScapes (0.1197). This indicates that MTSAM exhibits zero-shot capability when the domain gap is not excessively large, which aligns with your observation about the similarity between CityScapes and KITTI.

---

> ### Author Response · Authors · 2024-11-21
> **Response to Reviewer Zgv2 2/2**
>
> > Q4. Table 3 is slightly disappointing with LoRA-STL (r=32) beating MTSAM on some tasks. Not a show-stopper since overall performance seems ok.
>
> Table 3 shows the results on *PASCAL-Context* dataset. Compared to LoRA-STL (r=32), MTSAM achieves **2.53% improvement** on average with a lower number of trainable parameters. Specifically, MTSAM achieves significantly better performance in the semantic segmentation task and comparable results in the other three tasks. The slight performance decline in the three tasks compared to LoRA-STL is possibly due to conflicts in shared parameters among different tasks or biases from dominant tasks during model training. A feasible solution is to apply optimization algorithms from multi-task learning, e.g., loss-balance methods and gradient-balance methods. We consider this as an interesting direction for our future work.
>
>
> > Q5. The qualitative examples need to be better. Resolution is very low and hard to tell. I know cityscapes are like that, but perhaps testing on some higher res samples would be more convincing.
>
> Thanks for your valuable suggestion. Based on your suggestion, we tested MTSAM trained on the *CityScapes* dataset on high-resolution images, with results included in Appendix D of the updated manuscript. As can be seen, MTSAM outperforms other baselines across various tasks. In areas highlighted by white boxes, MTSAM generates more accurate results, demonstrating the effectiveness of MTSAM.

---

> ### Comment · Reviewer_Zgv2 · 2024-11-28
>
> I see.
>
> I will finalize my rating at 6. In my opinion, citiscapes and kitti are not zero shot at all, as they are very similar, so there is still question in my about the effectiveness, with such a 5-6 times drop in performance. NYU and cityscapes are completely different so it is hard to use them to gauge what is considered large drop.
>
> I still think your paper has merits, and also I have read the other reviewers' discussions with you. I think rating of 6 is a fair score.
>
> Thank you for taking time to respond to all my questions.

---

> > ### Author Response · Authors · 2024-11-28
> >
> > We appreciate your positive feedback on our paper. We would like to clarify that there exists a domain gap between the CityScapes and KITTI datasets due to the following reasons:
> > 1. The CityScapes dataset covers a larger distance range than the KITTI dataset [r1].
> > 2. Due to differences in their sources, such as the cameras used to collect data, the  KITTI (source) and CityScapes (target) datasets have been employed to measure performance in scenarios of unsupervised domain adaptation [r2].
> >
> > Hence, while both datasets are outdoor-based, they can still be considered to possess a certain domain shift, which makes them suitable for evaluating the zero-shot learning capabilities of a model.
> >
> > Moreover, the proposed MTSAM framework focuses on multi-task fine-tuning for multiple downstream tasks. It is not specifically designed to enhance zero-shot learning capabilities. Therefore, its performance in zero-shot scenarios may not be outstanding. However, based on the quantitative and qualitative experiments conducted, we believe that MTSAM, benefiting from the capabilities of SAM itself, exhibits certain zero-shot learning abilities.
> >
> > Finally, we thank you again for your timely response and positive feedback on our paper. We hope that our response addresses your concerns and we will appropriately revise the manuscript based on your suggestions.
> >
> > [r1] The Cityscapes Dataset for Semantic Urban Scene Understanding, CVPR 2016.
> >
> > [r2] Seeking Similarities over Differences: Similarity-based Domain Alignment for Adaptive Object Detection, ICCV 2021.

---

> > > ### Comment · Reviewer_Zgv2 · 2024-11-28
> > >
> > > Understood.
> > >
> > > Judging everything plus other reviewers' comments, I think 6 rating is a fair score, so I stand at that.
> > >
> > > Thanks.

---

### Official Review · Reviewer_SfrS · 2024-11-01

**Soundness:** 2
**Presentation:** 2
**Contribution:** 2
**Rating:** 6
**Confidence:** 3

**Summary:**

The paper proposes MTSAM, a multi-task learning framework that adapts the Segment Anything Model (SAM) for simultaneous execution of different computer vision tasks. By modifying SAM’s architecture and introducing a novel Tensorized low-Rank Adaptation (ToRA) method, MTSAM aims to optimize SAM's capabilities for multi-task scenarios, addressing task-specific output generation and efficient fine-tuning.

**Strengths:**

1.	The MTSAM framework successfully extends SAM’s architecture, enabling it to handle multiple downstream tasks, a significant improvement over SAM’s single-task limitations.
2.	ToRA offers a parameter-efficient fine-tuning solution that balances task-specific and shared information, enhancing performance without excessive resource requirements.
3.	Comprehensive experiments across benchmark datasets (NYUv2, CityScapes, and PASCAL-Context) demonstrate MTSAM's superior performance over existing methods in multi-task learning, indicating practical efficacy.
4.	The theoretical analysis of ToRA’s expressive power is well-presented and aligns with empirical findings, adding depth to the framework’s academic contributions.

**Weaknesses:**

1.	The first and second points in the summary of contribution points appear to be similar, the author is requested to provide good reasons for the significant difference, otherwise it is recommended to merge.
2.	It is hard for me to consider MTSAM as a substantial model innovation improvement, (1) I don't get why the prompt encoder affects the downstream adaptation of the model, (2) the author's improvement of the decoder is more like the integration of multiple decoders for different tasks.
3.	I am interested in ToRA, but the author's description of the details is unclear. (1) How to constrain U1, U2, and U3 to learn task-relevant and task-irrelevant information, respectively? (explain or experiment) (2) Why is the decomposition into U1+U2+U3+G just enough? What happens if there are more or less U's? What if there were G?
4.	The proposed ToRA lacks comparison with other LoRA-based improvement methods.

**Questions:**

See weakness.

---

> ### Author Response · Authors · 2024-11-21
> **Response to Reviewer SfrS 1/2**
>
> We sincerely thank you for providing valuable comments. You can find our response below for your concerns. Please kindly let us know if you have any further concerns.
>
> > Q1. The first and second points in the summary of contribution points appear to be similar, the author is requested to provide good reasons for the significant difference, otherwise it is recommended to merge.
>
> Thanks for your valuable suggestions. In our revised manuscript, we merge the first and second points in the summary of contributions points. Thus, the contributions of this paper are three-fold:
>
> * We propose MTSAM, a novel multi-task learning framework that extends the capabilities of SAM to perform multi-task learning. Specifically, we modify the original architecture of SAM by removing the prompt encoder and adding task embeddings. This modification enhances the flexibility of the original SAM.
> * We introduce ToRA, a novel multi-task PEFT method, that applies low-rank decomposition to the update parameter tensor, effectively learning both task-shared and task-specific information simultaneously, with theoretical analysis for its strong expressive power.
> * We conduct comprehensive experiments on benchmark datasets, demonstrating the exceptional performance of the proposed MTSAM framework.
>
> > Q2. It is hard for me to consider MTSAM as a substantial model innovation improvement,
> (1) I don't get why the prompt encoder affects the downstream adaptation of the model,
> (2) the author's improvement of the decoder is more like the integration of multiple decoders for different tasks.
>
>
> (1) In the SAM architecture, the prompt encoder extracts features from prompts (i.e., points, boxes, and masks), which are then used in cross-attention to compute the final segmentation masks. However, in a multi-task learning scenario for dense prediction tasks, we do not have prompts, so this part of the architecture cannot work during model inference. Therefore, we have removed it.
>
> (2) As mentioned in Figure 1 of the manuscript, applying SAM to multi-task learning faces the challenge of varying numbers of output channels for different tasks. For example, the number of output channels of the semantic segmentation task equals the number of classes, while that for the depth estimation task is always equal to 1. To address this challenge, we introduced task embedding, allowing the model to generate appropriate outputs according to task requirements while maintaining a unified architecture.
>
> | Method | Param. (M) $\downarrow$ | $\Delta_b$ $\uparrow$ |
> | -------- | -------- | -------- |
> | MLP             | 65.66     | +17.35%     |
> | Task Embedding  | 59.59     | +23.93%     |
>
> Moreover, to verify the effectiveness of our proposed task embedding, we attempted to adjust the output dimensions of the MLP in the decoder to achieve the desired output. As shown in the above table, this approach is less effective compared to using task embedding.
>
> > Q3. I am interested in ToRA, but the author's description of the details is unclear.
> (1) How to constrain U1, U2, and U3 to learn task-relevant and task-irrelevant information, respectively? (explain or experiment)
> (2) Why is the decomposition into U1+U2+U3+G just enough? What happens if there are more or less U's? What if there were G?
>
> (1) For the three-mode update parameter tensor $\Delta \mathbf{W}$, the first mode represents the output feature dimension, the second mode denotes the input feature dimension, and the third mode is for the task dimension. Hence, according to Tucker decomposition [r1], $U_1$ and $U_2$ reflect the main subspace variation of task-shared information corresponding to the first two modes in $\Delta \mathbf{W}$, while $U_3$ reflects the task-specific subspace structure corresponding to the last mode of $\Delta \mathbf{W}$. We make it clearer in the revision.
>
> (2) The number of $U$'s is determined by the mode of tensor [r1]. As the update parameter tensor $\Delta \mathbf{W} \in \mathbb{R}^{d \times k \times T}$ is a 3-mode tensor, Tucker decomposes it into three factor matrices $U_1$, $U_2$, $U_3$, and a 3-mode core tensor $\mathcal{G}$.
>
> [r1] Some mathematical notes on three-mode factor analysis. Psychometrika.

---

> ### Author Response · Authors · 2024-11-21
> **Response to Reviewer SfrS 2/2**
>
> > Q4. The proposed ToRA lacks comparison with other LoRA-based improvement methods.
>
> Thank you for your valuable suggestions. We have included a comparison with MultiLoRA [r2] in our experiments. Moreover, we added the comparisons with the latest published LoRA-based methods, Terra [r3] and HydraLoRA [r4]. The results are shown in the following table, and the complete results can be found in Table 7 of the updated manuscript. As can be seen, ToRA achieves better performance compared to those LoRA-based methods.
>
> | Method | Param. (M) $\downarrow$ | $\Delta_b$ $\uparrow$ |
> | -------- | -------- | -------- |
> | MultiLoRA [r2] | 65.12     | +20.11%     |
> | Terra [r3]     | 52.86     | +13.70%     |
> | HydraLoRA [r4] | 71.30     | +22.11%     |
> | ToRA           | 59.59     | +23.93%     |
>
>
> [r2] Multilora: Democratizing lora for better multi-task learning. arXiv preprint arXiv:2311.11501 2023.
>
> [r3] Time-Varying LoRA: Towards Effective Cross-Domain Fine-Tuning of Diffusion Models. NeurIPS 2024.
>
> [r4] HydraLoRA: An Asymmetric LoRA Architecture for Efficient Fine-Tuning. NeurIPS 2024.

---

> > ### Comment · Reviewer_SfrS · 2024-11-26
> >
> > Thanks to the author's response, which addressed most of my concerns.
> > Therefore, I can raise my score to 6 [marginally above the acceptance threshold].

---

> > > ### Author Response · Authors · 2024-11-26
> > >
> > > I appreciate your thoughtful review and feedback. Thank you for reconsidering my work and raising the score.

---

### Official Review · Reviewer_Sjqm · 2024-11-03

**Soundness:** 3
**Presentation:** 3
**Contribution:** 2
**Rating:** 6
**Confidence:** 4

**Summary:**

This paper propose the MTSAM, a multi-task segmentation model, which is based on the architecture of SAM. The researchers change the original prompt encoder and mask encoder of SAM into separate mask decoders for each downstream task, and introduce task embeddings to generate outputs with the corresponding number of channels, enabling the model can be adapted to various tasks. Otherwise, the researchers apply a low-rank tensor decomposition method to fine-tune the image encoder of MTSAM. different tasks. The proposed ToRA can use both task-shared and task-specific information during the multi-task fine-tuning process. The experimental results demonstrate the effectiveness of MTSAM.

**Strengths:**

There are several strengths of the paper:
For originality, based on the original model structure of SAM, this work makes a simple, direct but effective modification. Inspired by the previous work, it also proposes the Tora, a novel multi-task PEFT method on the idea of low-rank tensor decomposition. It has made effective innovations on the basis of existing work.
For clarity, the writing of the article is smooth, and the formulas, figures, etc. are clear and unambiguous.
For quality, This work was carried out on the NYUv2, CityScapes, and PASCAL-Context, and the experiments are fairly convincing in terms of the results of the indicators presented.
For significance, this work migrates the excellent performance of SAM to multiple downstream tasks, which has certain significance for the further promotion of SAM.

**Weaknesses:**

The weakness of this paper lies in the following aspects:
1.The comparative advantages primarily focus on CNN-based methods, without analyzing more advanced approaches like SwinSTL. Additionally, the performance metrics compared to SwinSTL are not significantly superior.
2.The dataset metric configurations differ without adequate explanation. For instance, it is unclear why different evaluation metrics are applied to segmentation tasks across various datasets.
3.The results section insufficiently demonstrates the effectiveness of image multitasking, and the supplementary appendix provides limited image results.

**Questions:**

1.Could you provide a more detailed analysis of the advantages over advanced methods like SwinSTL?
2.Can you clarify why different evaluation metrics are used for segmentation tasks or others across various datasets? What criteria were used to select these metrics?
3.Would it be possible to include more comprehensive examples of image multitasking in the results section?

---

> ### Author Response · Authors · 2024-11-21
> **Response to Reviewer Sjqm**
>
> We sincerely thank you for providing valuable comments. You can find our response below for your concerns. Please kindly let us know if you have any further concerns.
>
> > Q1. Could you provide a more detailed analysis of the advantages over advanced methods like SwinSTL?
>
> Compared to Swin-based architectures (i.e., VTAGML and SwinMTL), we propose **a novel multi-task parameter-efficient fine-tuning framework**, MTSAM, which effectively leverages the rich semantic knowledge in the foundation model SAM. Specifically, MTSAM offers the following advantages:
> * MTSAM achieves **better performance** compared to those baselines. Specifically, MTSAM shows 4.38% and 12.45% improvement over SwinMTL on the *NYUv2* and *CityScapes* datasets, respectively.
> * MTSAM demonstrates **better parameter efficiency**, offering advantages in storage and enhancing its practical application value.
>
>
> > Q2. Can you clarify why different evaluation metrics are used for segmentation tasks or others across various datasets? What criteria were used to select these metrics?
>
> We **follow the classic setups of [r1,r2]** to evaluate the performance of different tasks on different datasets. The evaluation metric of each task has been introduced in Appendix B.1.
>
> * On *NYUv2* and *CityScapes* datasets:
>     * For the semantic segmentation task, we use mIoU and Pixel Accuracy (Pix Acc).
>     * For the depth prediction task, we use Absolute Error (Abs Err) and Relative Error (Rel Err).
>     * For the surface normal estimation task, we evaluate the mean and median of angular errors measured in degrees and the percentage of pixels with angular errors within 11.25°, 22.5°, and 30°.
> * On *PASCAL-Context* dataset:
>     * We use mIoU to evaluate the semantic segmentation, human parts segmentation, and saliency estimation tasks.
>     * For the surface normal task, we use the mean of angular errors measured in degrees.
>
> [r1] Polyhistor: Parameter-efficient multi-task adaptation for dense vision tasks. NeurIPS 2022.
>
> [r2] End-to-end multi-task learning with attention. CVPR 2019.
>
> > Q3. Would it be possible to include more comprehensive examples of image multitasking in the results section?
>
> Thanks for your valuable suggestions. In our revised manuscript, we have added more qualitative results in Appendix D to provide comprehensive examples of image multitasking.

---

> > ### Comment · Reviewer_Sjqm · 2024-11-26
> >
> > Thanks to the author's response, which addressed most of my concerns. Therefore, I can raise my score to 6 [marginally above the acceptance threshold].

---

> > > ### Author Response · Authors · 2024-11-26
> > >
> > > I appreciate your thoughtful review and feedback. Thank you for reconsidering my work and raising the score.

---

### Official Review · Reviewer_zxBa · 2024-11-03

**Soundness:** 3
**Presentation:** 3
**Contribution:** 2
**Rating:** 6
**Confidence:** 4

**Summary:**

The paper proposes MTSAM (Multi-Task SAM), a framework that extends the Segment Anything Model (SAM) for multi-task learning. SAM's original architecture is limited to single-task applications due to its prompt encoder and uniform output channels. MTSAM addresses these limitations by modifying SAM’s architecture to accommodate task-specific outputs and by introducing Tensorized low-Rank Adaptation (ToRA) for multi-task fine-tuning. ToRA injects a tensor parameter into SAM’s encoder, allowing efficient handling of both shared and task-specific information. Extensive experiments on benchmark datasets (NYUv2, CityScapes, PASCAL-Context) show that MTSAM outperforms existing multi-task learning approaches, both qualitatively and quantitatively, in segmentation, depth estimation, and surface normal prediction.

**Strengths:**

1) MTSAM successfully extends SAM to a multi-task framework, a novel approach that leverages SAM's strong zero-shot capabilities in a multi-task setting.

2) The proposed ToRA method is parameter-efficient, enabling sublinear parameter growth and effective use of shared information across tasks.

3) The paper provides theoretical justification for ToRA’s superiority over existing methods like LoRA, adding credibility to its parameter efficiency claims.

4) MTSAM outperforms other multi-task learning models across three benchmark datasets, demonstrating its efficacy in varied visual tasks.

**Weaknesses:**

1) The current work lacks an evaluation of MTSAM’s zero-shot generalization ability, particularly on unseen data distributions.

2) The experiments do not include a direct comparison with full fine-tuning methods, leaving it unclear whether MTSAM’s parameter-efficient fine-tuning can achieve competitive performance without compromising accuracy.

3) There is a lack of comparative experiments to demonstrate the effectiveness of task embedding. Further experiments are needed to confirm the specific advantages of task embedding in multi-task setups.

**Questions:**

I don't have particular questions. The concerns can be found on the weaknesses section.

---

> ### Author Response · Authors · 2024-11-21
> **Response to Reviewer zxBa**
>
> We sincerely thank you for providing valuable comments. You can find our response below for your concerns. Please kindly let us know if you have any further concerns.
>
> > Q1. The current work lacks an evaluation of MTSAM’s **zero-shot generalization ability**, particularly on unseen data distributions.
>
> First, we would like to humbly clarify that our proposed framework, MTSAM, is primarily designed for **efficient multi-task fine-tuning of SAM**, rather than for addressing out-of-distribution data scenarios.
>
> Nonetheless, to evaluate its performance on unseen data as suggested, we applied the model fine-tuned on the *NYUv2* dataset to make depth predictions on the *CityScapes* dataset. Qualitative results are provided in Figure 12 of Appendix E, and illustrate that **MTSAM is capable of handling unseen data distributions** to some extent.
>
>
> > Q2. The experiments do not include a direct **comparison with full fine-tuning methods**, leaving it unclear whether MTSAM’s parameter-efficient fine-tuning can achieve competitive performance without compromising accuracy.
>
> Thanks for your valuable suggestions. We conducted a comparison with the full fine-tuning method on the *NYUv2* dataset. As shown in the following table, the results demonstrate that MTSAM achieves significant improvements over the full fine-tuning method. This improvement is attributed to MTSAM’s high parameter and sample efficiency in multi-task parameter-efficient tuning, whereas full fine-tuning requires a larger number of samples to converge and achieve the same level of performance.
>
> | Method | Param. (M) $\downarrow$ | $\Delta_b$ $\uparrow$ |
> | -------- | -------- | -------- |
> | Full fine-tuning | 1222.47   | +14.57%     |
> | MTSAM            | 59.59     | +23.93%     |
>
> As suggested, the result of full fine-tuning method has been added in Appendix C.1 of the revised manuscript.
>
>
> > Q3. There is a lack of comparative experiments to demonstrate **the effectiveness of task embedding**. Further experiments are needed to confirm the specific advantages of task embedding in multi-task setups.
>
> To demonstrate the effectiveness of the proposed task embedding, we compared it with the method of modifying the MLP output dimensions for different tasks on the *NYUv2* dataset. As shown in the table below, **task embedding performs better**.
> This improvement is due to the interaction between task embeddings and image features through the cross-attention mechanism, which enables the decoder to better learn the task-specific knowledge and achieve superior results. Those results and analyses have been added in Appendix C.2 of the revised manuscript.
>
> | Method | Param. (M) $\downarrow$ | $\Delta_b$ $\uparrow$ |
> | -------- | -------- | -------- |
> | MLP             | 65.66     | +17.35%     |
> | Task Embedding  | 59.59     | +23.93%     |

---

### Author Response · Authors · 2024-12-03
**General Response**

Dear Area Chairs and Reviewers,

As the authors of this paper, we are glad to have this opportunity to express the current position of our paper.

We thank all reviewers for taking the time to review our work and giving us constructive and valuable comments to improve the paper. All the reviewers provided positive feedback.
1. All the reviewers agreed on the novelty of the proposed MTSAM framework (Reviewers `zxBa`, `Sjqm`, `SfrS`, and `Zgv2`), as it effectively addresses the single-task limitation of SAM (Reviewers `SfrS` and `Zgv2`) and has practical significance for SAM's broader application (Reviewer `Sjqm`).
2. Reviewers also praised the parameter efficiency and effectiveness of the ToRA method (Reviewers `zxBa`, `SfrS`, and `Zgv2`).
3. Additionally, reviewers appreciated the theoretical justification provided for ToRA's superiority over existing methods, which adds depth to the contribution of the paper (Reviewers `zxBa` and `SfrS`).
4. Moreover, reviewers commended the writing and confirmed the effectiveness of the proposed framework in the experiments (Reviewers `zxBa`, `Sjqm`, and `SfrS`).

During the rebuttal period, we responded to all the comments of all the reviewers and revised the paper accordingly (highlighted in blue).

Thank you once again for your kind consideration of our work.

Best regards,

Authors

---

### Meta-Review · Area_Chair_Z5kz · 2024-12-20

**Metareview:**

This work leverages SAM for multi-task learning. Specifically, the architecture of SAM is modified to enable the generation of task-specific outputs. In addition, an efficient fine-tuning strategy, i.e., Tensorized low-Rank Adaptation (ToRA), is proposed to train multi-task SAM effectively. The initial scores were mixed and major concerns were about performance and ablation study, evaluation metrics, and better demonstration. Most of concerns were addressed by rebuttal and all reviewers had positive scores. Please include the experiments and comments from the discussion in the revised submission.

**Additional Comments On Reviewer Discussion:**

Reviewer Sjqm had concerns about performance and evaluation, which were addressed by rebuttal. Therefore, the rating of the reviewer is increased to 6. In addition, concerns from Reviewer SfrS about novelty and ablation study were also discussed and led to a better score. While Reviewer Zgv2 kept the original positive score, the merits of the work were confirmed.

---

### Decision · Program_Chairs · 2025-01-22

Accept (Poster)